# Fast Convergence of Natural Gradient Descent for Overparameterized Neural Networks

**Guodong Zhang**[1,2], **James Martens**[3], **Roger Grosse**[1,2]
University of Toronto[1], Vector Institute[2], DeepMind[3]
{gdzhang, rgrosse}@cs.toronto.edu, jamesmartens@google.com

## Abstract

Natural gradient descent has proven effective at mitigating the effects of pathological curvature in neural network optimization, but little is known theoretically about its convergence properties, especially for *nonlinear* networks. In this work, we analyze for the first time the speed of convergence of natural gradient descent on nonlinear neural networks with squared-error loss. We identify two conditions which guarantee efficient convergence from random initializations: (1) the Jacobian matrix (of network's output for all training cases with respect to the parameters) has full row rank, and (2) the Jacobian matrix is stable for small perturbations around the initialization. For two-layer ReLU neural networks, we prove that these two conditions do in fact hold throughout the training, under the assumptions of nondegenerate inputs and overparameterization. We further extend our analysis to more general loss functions. Lastly, we show that K-FAC, an approximate natural gradient descent method, also converges to global minima under the same assumptions, and we give a bound on the rate of this convergence.

## 1 Introduction

Because training large neural networks is costly, there has been much interest in using second-order optimization to speed up training [Becker and LeCun, 1989, Martens, 2010, Martens and Grosse, 2015], and in particlar natural gradient descent [Amari, 1998, 1997]. Recently, scalable approximations to natural gradient descent have shown practical success in a variety of tasks and architectures [Martens and Grosse, 2015, Grosse and Martens, 2016, Wu et al., 2017, Zhang et al., 2018a, Martens et al., 2018]. Natural gradient descent has an appealing interpretation as optimizing over a Riemannian manifold using an intrinsic distance metric; this implies the updates are invariant to transformations such as whitening [Ollivier, 2015, Luk and Grosse, 2018]. It is also closely connected to Gauss-Newton optimization, suggesting it should achieve fast convergence in certain settings [Pascanu and Bengio, 2013, Martens, 2014, Botev et al., 2017].

Does this intuition translate into faster convergence? Amari [1998] provided arguments in the affirmative, as long as the cost function is well approximated by a convex quadratic. However, it remains unknown whether natural gradient descent can optimize neural networks faster than gradient descent — a major gap in our understanding. The problem is that the optimization of neural networks is both nonconvex and non-smooth, making it difficult to prove nontrivial convergence bounds. In general, finding a global minimum of a general non-convex function is an NP-complete problem, and neural network training in particular is NP-complete [Blum and Rivest, 1992].

However, in the past two years, researchers have finally gained substantial traction in understanding the dynamics of gradient-based optimization of neural networks. Theoretically, it has been shown that gradient descent starting from a random initialization is able to find a global minimum if the network is wide enough [Li and Liang, 2018, Du et al., 2018b,a, Zou et al., 2018, Allen-Zhu et al., 2018, Oymak and Soltanolkotabi, 2019]. The key technique of those works is to show that neural

networks become well-behaved if they are largely overparameterized in the sense that the number of hidden units is polynomially large in the size of the training data. However, most of these works have focused on standard gradient descent, leaving open the question of whether similar statements can be made about other optimizers.

Most convergence analysis of natural gradient descent has focused on simple convex quadratic objectives (e.g. [Martens, 2014]). Very recently, the convergence properties of NGD were studied in the context of linear networks [Bernacchia et al., 2018]. While the linearity assumption simplifies the analysis of training dynamics [Saxe et al., 2013], linear networks are severely limited in terms of their expressivity, and it's not clear which conclusions will generalize from linear to nonlinear networks.

In this work, we analyze natural gradient descent for *nonlinear networks*. We give two simple and generic conditions on the Jacobian matrix which guarantee efficient convergence to a global minimum. We then apply this analysis to a particular distribution over two-layer ReLU networks which has recently been used to analyze the convergence of gradient descent [Li and Liang, 2018, Du et al., 2018a, Oymak and Soltanolkotabi, 2019]. We show that for sufficiently high network width, NGD will converge to the global minimum. We give bounds on the convergence rate of two-layer ReLU networks that are much better than the analogous bounds that have been proven for gradient descent [Du et al., 2018b, Wu et al., 2019, Oymak and Soltanolkotabi, 2019], while allowing for much higher learning rates. Moreover, in the limit of infinite width, and assuming a squared error loss, we show that NGD converges in just *one iteration*. The main contributions of our work are summarized as follows:

- We provide the first convergence result for natural gradient descent in training randomly-initialized overparameterized neural networks where the number of hidden units is polynomially larger than the number of training samples. We show that natural gradient descent gives an $\mathcal{O}(\lambda_{\min}(\mathbf{G}^\infty))$ improvement in convergence rate given the same learning rate as gradient descent, where $\mathbf{G}^\infty$ is a Gram matrix that depends on the data.
- We show that natural gradient enables us to use a much larger step size, resulting in an even faster convergence rate. Specifically, the maximal step size of natural gradient descent is $\mathcal{O}(1)$ for (polynomially) wide networks.
- We show that K-FAC [Martens and Grosse, 2015], an approximate natural gradient descent method, also converges to global minima with linear rate, although this result requires a higher level of overparameterization compared to GD and exact NGD.
- We analyze the generalization properties of NGD, showing that the improved convergence rates *provably* don't come at the expense of worse generalization.

## 2 Related Works

Recently, there have been many works studying the optimization problem in deep learning, i.e., why in practice many neural network architectures reliably converge to global minima (zero training error). One popular way to attack this problem is to analyze the underlying loss surface [Hardt and Ma, 2016, Kawaguchi, 2016, Kawaguchi and Bengio, 2018, Nguyen and Hein, 2017, Soudry and Carmon, 2016]. The main argument of those works is that there are no bad local minima. It has been proven that gradient descent can find global minima [Ge et al., 2015, Lee et al., 2016] if the loss surface satisfies: (1) all local minima are global and (2) all saddle points are strict in the sense that there exists at least one negative curvature direction. Unfortunately, most of those works rely on unrealistic assumptions (e.g., linear activations [Hardt and Ma, 2016, Kawaguchi, 2016]) and cannot generalize to practical neural networks. Moreover, Yun et al. [2018] shows that small nonlinearity in shallow networks can create bad local minima.

Another way to understand the optimization of neural networks is to directly analyze the optimization dynamics. Our work also falls within this category. However, most work in this direction focuses on gradient descent. Bartlett et al., Arora et al. [2019a] studied the optimization trajectory of deep linear networks and showed that gradient descent can find global minima under some assumptions. Previously, the dynamics of linear networks have also been studied by Saxe et al. [2013], Advani and Saxe [2017]. For nonlinear neural networks, a series of papers [Tian, 2017, Brutzkus and Globerson, 2017, Du et al., 2017, Li and Yuan, 2017, Zhang et al., 2018b] studied a specific class of shallow two-layer neural networks together with strong assumptions on input distribution as well as realizability of labels, proving global convergence of gradient descent. Very recently, there are

some works proving global convergence of gradient descent [Li and Liang, 2018, Du et al., 2018b,a, Allen-Zhu et al., 2018, Zou et al., 2018, Gao et al., 2019] or adaptive gradient methods [Wu et al., 2019] on overparameterized neural networks. More specifically, Li and Liang [2018], Allen-Zhu et al. [2018], Zou et al. [2018] analyzed the dynamics of weights and showed that the gradient cannot be small if the objective value is large. On the other hand, Du et al. [2018b,a], Wu et al. [2019] studied the dynamics of the outputs of neural networks, where the convergence properties are captured by a Gram matrix. Our work is very similar to Du et al. [2018b], Wu et al. [2019]. We note that these papers all require the step size to be sufficiently small to guarantee the global convergence, leading to slow convergence.

To our knowledge, there is only one paper [Bernacchia et al., 2018] studying the global convergence of natural gradient for neural networks. However, Bernacchia et al. [2018] only studied deep linear networks with infinitesimal step size and squared error loss functions. In this sense, our work is the first one proving global convergence of natural gradient descent on nonlinear networks.

There have been many attempts to understand the generalization properties of neural networks since Zhang et al. [2016]'s seminal paper. Researchers have proposed norm-based generalization bounds [Neyshabur et al., 2015, 2017, Bartlett and Mendelson, 2002, Bartlett et al., 2017, Golowich et al., 2017], compression bounds [Arora et al., 2018] and PAC-Bayes bounds [Dziugaite and Roy, 2017, 2018, Zou et al., 2018]. Recently, overparameterization of neural networks together with good initialization has been believed to be one key factor of good generalization. Neyshabur et al. [2019] empirically showed that wide neural networks stay close to the initialization, thus leading to good generalization. Theoretically, researchers did prove that overparameterization as well as linear convergence jointly restrict the weights to be close to the initialization [Du et al., 2018b,a, Allen-Zhu et al., 2018, Zou et al., 2018, Arora et al., 2019b]. The most closely related paper is Arora et al. [2019b], which shows that the optimization and generalization phenomenon can be explained by a Gram matrix. The main difference is that our analysis is based on natural gradient descent, which converges faster and provably generalizes as well as gradient descent.

Concurrently and independently, Cai et al. [2019] showed that natural gradient descent (they call it Gram-Gauss-Newton) enjoys quadratic convergence rate guarantee for overparameterized networks on regression problems. Additionally, they showed that it is much cheaper to precondition the gradient in the output space when the number of data points is much smaller than the number of parameters.

## 3 Convergence Analysis of Natural Gradient Descent

We begin our convergence analysis of natural gradient descent – under appropriate conditions – for the neural network optimization problem. Formally, we consider a generic neural network $f(\boldsymbol{\theta}, \mathbf{x})$ with a single output and squared error loss $\ell(u, y) = \frac{1}{2}(u - y)^2$ for simplicity[1], where $\boldsymbol{\theta} \in \mathbb{R}^m$ denots all parameters of the network (i.e. weights and biases). Given a training dataset $\{(\mathbf{x}_i, y_i)\}_{i=1}^n$, we want to minimize the following loss function:

$$\mathcal{L}(\boldsymbol{\theta}) = \frac{1}{n} \sum_{i=1}^{n} \ell\left(f(\boldsymbol{\theta}, \mathbf{x}_i), y_i\right) = \frac{1}{2n} \sum_{i=1}^{n} \left(f(\boldsymbol{\theta}, \mathbf{x}_i) - y_i\right)^2. \tag{1}$$

One main focus of this paper is to analyze the following procedure:

$$\boldsymbol{\theta}(k + 1) = \boldsymbol{\theta}(k) - \eta \mathbf{F}(\boldsymbol{\theta}(k))^{-1} \frac{\partial \mathcal{L}(\boldsymbol{\theta}(k))}{\partial \boldsymbol{\theta}(k)}, \tag{2}$$

where $\eta > 0$ is the step size, and $\mathbf{F}$ is the Fisher information matrix associated with the network's predictive distribution over $y$ (which is implied by its loss function and is $\mathcal{N}(f(\boldsymbol{\theta}, \mathbf{x}_i), 1)$ for the squared error loss) and the dataset's distribution over $\mathbf{x}$.

As shown by Martens [2014], the Fisher $\mathbf{F}$ is equivalent to the generalized Gauss-Newton matrix, defined as $\mathbb{E}_{\mathbf{x}_i}\left[\mathbf{J}_i^{\top} \mathbf{H}_\ell \mathbf{J}_i\right]$ if the predictive distribution is in the exponential family, such as categorical distribution (for classification) or Gaussian distribution (for regression). $\mathbf{J}_i$ is the Jacobian matrix of $\mathbf{u}_i$ with respect to the parameters $\boldsymbol{\theta}$ and $\mathbf{H}_\ell$ is the Hessian of the loss $\ell(\mathbf{u}, \mathbf{y})$ with respect to the network prediction $\mathbf{u}$ (which is $\mathbf{I}$ in our setting). Therefore, with the squared error loss, the Fisher

matrix can be compactly written as $\mathbf{F} = \mathbb{E}\left[\mathbf{J}_i^\top \mathbf{J}_i\right] = \frac{1}{n}\mathbf{J}^\top\mathbf{J}$ (which coincides with classical Gauss-Newton matrix), where $\mathbf{J} = [\mathbf{J}_1^\top, ..., \mathbf{J}_n^\top]^\top$ is the Jacobian matrix for the whole dataset. In practice, when the number of parameters $m$ is larger than number of samples $n$ we have, the Fisher information matrix $\mathbf{F} = \frac{1}{n}\mathbf{J}^\top\mathbf{J}$ is surely singular. In that case, we take the generalized inverse [Bernacchia et al., 2018] $\mathbf{F}^\dagger = n\mathbf{J}^\top\mathbf{G}^{-1}\mathbf{G}^{-1}\mathbf{J}$ with $\mathbf{G} = \mathbf{J}\mathbf{J}^\top$, which gives the following update rule:

$$\boldsymbol{\theta}(k+1) = \boldsymbol{\theta}(k) - \eta\mathbf{J}^\top\left(\mathbf{J}\mathbf{J}^\top\right)^{-1}(\mathbf{u} - \mathbf{y}), \tag{3}$$

where $\mathbf{u} = [\mathbf{u}_1, ..., \mathbf{u}_n]^\top = [f(\boldsymbol{\theta}, \mathbf{x}_1), ..., f(\boldsymbol{\theta}, \mathbf{x}_n)]^\top$ and $\mathbf{y} = [y_1, ..., y_n]^\top$.

We now introduce two conditions on the network $f_{\boldsymbol{\theta}}$ that suffice for proving the global convergence of NGD to a minimizer which achieves zero training loss (and is therefore a global minimizer). To motivate these two conditions we make the following observations. First, the global minimizer is characterized by the condition that the gradient in the output space is zero for each case (i.e. $\nabla_{\mathbf{u}}\mathcal{L}(\boldsymbol{\theta}) = \mathbf{0}$). Meanwhile, local minima are characterized by the condition that the gradient with respect to the parameters $\nabla_{\boldsymbol{\theta}}\mathcal{L}(\boldsymbol{\theta})$ is zero. Thus, one way to avoid finding local minima that aren't global is to ensure that the parameter gradient is zero if and only if the output space gradient (for each case) is zero. It's not hard to see that this property holds as long as $\mathbf{G}$ remains non-singular throughout optimization (or equivalently that $\mathbf{J}$ always has full row rank). The following two conditions ensure that this happens, by first requiring that this property hold at initialization time, and second that $\mathbf{J}$ changes slowly enough that it remains true in a big enough neighborhood around $\boldsymbol{\theta}(0)$.

**Condition 1** (Full row rank of Jacobian matrix). *The Jacobian matrix $\mathbf{J}(0)$ at the initialization has full row rank, or equivalently, the Gram matrix $\mathbf{G}(0) = \mathbf{J}(0)\mathbf{J}(0)^\top$ is positive definite.*

**Remark 1.** *Condition 1 implies that $m \leq n$, which means the Fisher information matrix is singular and we have to use the generalized inverse except in the case where $m = n$.*

**Condition 2** (Stable Jacobian). *There exists $0 \leq C < \frac{1}{2}$ such that for all parameters $\boldsymbol{\theta}$ that satisfy $\|\boldsymbol{\theta} - \boldsymbol{\theta}(0)\|_2 \leq \frac{3\|\mathbf{y} - \mathbf{u}(0)\|_2}{\sqrt{\lambda_{\min}(\mathbf{G}(0))}}$, we have*

$$\|\mathbf{J}(\theta) - \mathbf{J}(0)\|_2 \leq \frac{C}{3}\sqrt{\lambda_{\min}(\mathbf{G}(0))}. \tag{4}$$

This condition shares the same spirit with the Lipschtiz smoothness assumption in classical optimization theory. It implies (with small $C$) that the network is close to a linearized network [Lee et al., 2019] around the initialization and therefore natural gradient descent update is close to the gradient descent update in the output space. Along with Condition 1, we have the following theorem.

**Theorem 1** (Natural gradient descent). *Let Condition 1 and 2 hold. Suppose we optimize with NGD using a step size $\eta \leq \frac{1-2C}{(1+C)^2}$. Then for $k = 0, 1, 2, ...$ we have*

$$\|\mathbf{u}(k) - \mathbf{y}\|_2^2 \leq (1 - \eta)^k\|\mathbf{u}(0) - \mathbf{y}\|_2^2. \tag{5}$$

To be noted, $\|\mathbf{u}(k) - \mathbf{y}\|_2^2$ is the squared error loss up to a constant. Due to space constraints we only give a short sketch of the proof here. The full proof is given in Appendix B.

**Proof Sketch**. *Our proof relies on the following insights. First, if the Jacobian matrix has full row rank, this guarantees linear convergence for infinitesimal step size. The linear convergence property restricts the parameters to be close to the initialization, which implies the Jacobian matrix is always full row rank throughout the training, and therefore natural gradient descent with infinitesimal step size converges to global minima. Furthermore, given the network is close to a linearized network (since the Jacobian matrix is stable with respect to small perturbations around the initialization), we are able to extend the proof to discrete time with a large step size.*

In summary, we prove that NGD exhibits linear convergence to the global minimizer of the neural network training problem, under Conditions 1 and 2. We believe our arguments in this section are general (i.e., architecture-agnostic), and can serve as a recipe for proving global convergence of natural gradient descent in other settings.

## 3.1 Other Loss Functions

We note that our analysis can be easily extended to more general loss function class. Here, we take the class of functions that are $\mu$-strongly convex with $L$-Lipschitz gradients as an example. Note that

strongly convexity is a very mild assumption since we can always add $L_2$ regularization to make the convex loss strongly convex. Therefore, this function class includes regularized cross-entropy loss (which is typically used in classification) and squared error (for regression). For this type of loss, we need a strong version of Condition 2.

**Condition 3** (Stable Jacobian). *There exists $0 \leq C < \frac{1}{1+\kappa}$ such that for all parameters $\boldsymbol{\theta}$ that satisfy $\|\boldsymbol{\theta} - \boldsymbol{\theta}(0)\|_2 \leq \frac{3(1+\kappa)\|\mathbf{y}-\mathbf{u}(0)\|_2}{2\sqrt{\lambda_{\min}(\mathbf{G}(0))}}$ where $\kappa = \frac{L}{\mu}$*

$$\|\mathbf{J}(\theta) - \mathbf{J}(0)\|_2 \leq \frac{C}{3}\sqrt{\lambda_{\min}(\mathbf{G}(0))}. \tag{6}$$

**Theorem 2.** *Under Condition 1 and 3, but with $\mu$-strongly convex loss function $\ell(\cdot, \cdot)$ with $L$-Lipschitz gradient ($\kappa = \frac{L}{\mu}$), and we set the step size $\eta \leq \frac{2}{\mu+L}\frac{1-(1+\kappa)C}{(1+C)^2}$, then we have for $k = 0, 1, 2, ...$*

$$\|\mathbf{u}(k) - \mathbf{y}\|_2^2 \leq \left(1 - \frac{2\eta\mu L}{\mu + L}\right)^k \|\mathbf{u}(0) - \mathbf{y}\|_2^2. \tag{7}$$

The key step of proving Theorem 2 is to show if $m$ is large enough, then natural gradient descent is approximately gradient descent in the output space. Thus the results can be easily derived according to standard bounds for convex optimization. Due to the page limit, we defer the proof to the Appendix C.

**Remark 2.** *In Theorem 2, the convergence rate depends on the condition number $\kappa = \frac{L}{\mu}$, which can be removed if we take into the curvature information of the loss function. In other words, we expect that the bound has no dependency on $\kappa$ if we use the Fisher matrix rather than the classical Gauss-Newton (assuming Euclidean metric in the output space [Luk and Grosse, 2018]) in Theorem 2.*

## 4 Optimizing Overparameterized Neural Networks

In Section 3, we analyzed the convergence properties of natural gradient descent, under the abstract Conditions 1 and 2. In this section, we make our analysis concrete by applying it to a specific type of overparameterized network (with a certain random initialization). We show that Conditions 1 and 2 hold with high probability. We therefore establish that NGD exhibits linear convergence to a global minimizer for such networks.

### 4.1 Notation

We let $[m] = \{1, 2, ..., m\}$. We use $\otimes, \odot$ to denote the Kronecker and Hadamard products. And we use $*$ and $\star$ to denote row-wise and column-wise Khatri-Rao products, respectively. For a matrix $\mathbf{A}$, we use $\mathbf{A}_{ij}$ to denote its $(i, j)$-th entry. We use $\|\cdot\|_2$ to denote the Euclidean norm of a vector or spectral norm of a matrix and $\|\cdot\|_{\mathbf{F}}$ to denote the Frobenius norm of a matrix. We use $\lambda_{\max}(\mathbf{A})$ and $\lambda_{\min}(\mathbf{A})$ to denote the largest and smallest eigenvalue of a square matrix, and $\sigma_{\max}(\mathbf{A})$ and $\sigma_{\min}(\mathbf{A})$ to denote the largest and smallest singular value of a (possibly non-square) matrix. For a positive definite matrix $\mathbf{A}$, we use $\kappa_{\mathbf{A}}$ to denote its condition number, i.e., $\lambda_{\max}(\mathbf{A})/\lambda_{\min}(\mathbf{A})$. We also use $\langle \cdot, \cdot \rangle$ to denote the standard inner product between two vectors. Given an event $E$, we use $\mathbb{I}\{E\}$ to denote the indicator function for $E$.

### 4.2 Problem Setup

Formally, we consider a neural network of the following form:

$$f(\mathbf{w}, \mathbf{a}, \mathbf{x}) = \frac{1}{\sqrt{m}} \sum_{r=1}^m a_r \phi(\mathbf{w}_r^\top \mathbf{x}), \tag{8}$$

where $\mathbf{x} \in \mathbb{R}^d$ is the input, $\mathbf{w} = \left[\mathbf{w}_1^\top, ..., \mathbf{w}_r^\top\right]^\top \in \mathbb{R}^{md}$ is the weight matrix (formed into a vector) of the first layer, $a_r \in \mathbb{R}$ is the output weight of hidden unit $r$ and $\phi(\cdot)$ is the ReLU activation function (acting entry-wise for vector arguments). For $r \in [m]$, we initialize the weights of first layer $\mathbf{w}_r \sim \mathcal{N}(\mathbf{0}, \nu^2\mathbf{I})$ and output weight $a_r \sim \mathbf{unif}[\{-1, +1\}]$.

Following Du et al. [2018b], Wu et al. [2019], we make the following assumption on the data.

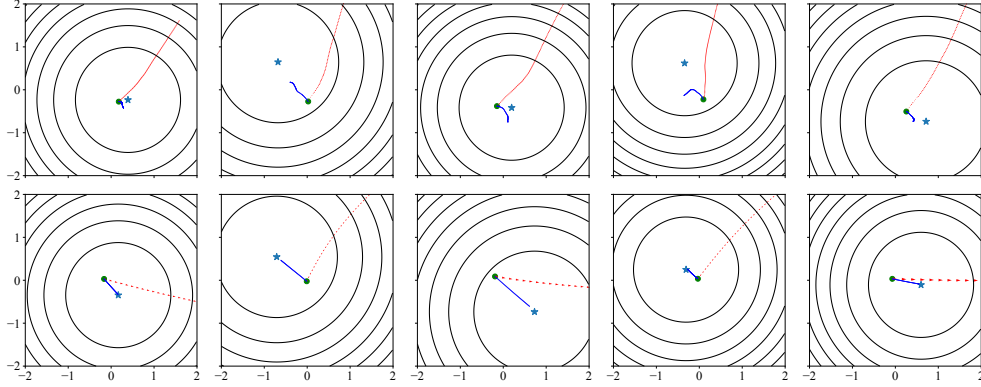

**Figure 1:** Visualization of natural gradient update and gradient descent update in the output space (for a randomly initialized network). We take two classes (4 and 9) from MNIST [LeCun et al., 1998] and generate the targets (denoted as star in the figure) by $f(x) = x - 0.5 + 0.3 \times \mathcal{N}(0, \mathbf{I})$ where $x \in \mathbb{R}^2$ is one-hot target. We get natural gradient update by running 100 iterations of conjugate gradient [Martens, 2010]. **The first row**: a MLP with two hidden layers and 100 hidden units in each layer. **The second row**: a MLP with two hidden layers and 6000 hidden units in each layer. In both cases, ReLU activation function was used. We interpolate the step size from 0 to 1. For the over-parameterized network (in the second row), natural gradient descent (implemented by conjugate gradient) matches output space gradient well.

**Assumption 1.** *For all $i$, $\|\mathbf{x}_i\|_2 = 1$ and $|y_i| = \mathcal{O}(1)$. For any $i \neq j$, $\mathbf{x}_i \nparallel \mathbf{x}_j$.*

This very mild condition simply requires the inputs and outputs have standardized norms, and that different input vectors are distinguishable from each other. Datasets that do not satisfy this condition can be made to do so via simple pre-processing.

Following Du et al. [2018b], Oymak and Soltanolkotabi [2019], Wu et al. [2019], we only optimize the weights of the first layer[2], i.e., $\boldsymbol{\theta} = \mathbf{w}$. Therefore, natural gradient descent can be simplified to

$$\mathbf{w}(k + 1) = \mathbf{w}(k) - \eta \mathbf{J}^\top (\mathbf{J}\mathbf{J}^\top)^{-1}(\mathbf{u} - \mathbf{y}). \tag{9}$$

Though this is only a shallow fully connected neural network, the objective is still non-smooth and non-convex [Du et al., 2018b] due to the use of ReLU activation function. We further note that this two-layer network model has been useful in understanding the optimization and generalization of deep neural networks [Xie et al., 2016, Li and Liang, 2018, Du et al., 2018b, Arora et al., 2019b, Wu et al., 2019], and some results have been extended to multi-layer networks [Du et al., 2018a].

Following Du et al. [2018b], Wu et al. [2019], we define the *limiting Gram matrix* as follows:

**Definition 1** (Limiting Gram Matrix). *The limiting Gram matrix $\mathbf{G}^\infty \in \mathbb{R}^{n \times n}$ is defined as follows. For $(i, j)$- entry, we have*

$$\mathbf{G}_{ij}^\infty = \mathbb{E}_{\mathbf{w} \sim \mathcal{N}(\mathbf{0}, \nu^2 \mathbf{I})} \left[ \mathbf{x}_i^\top \mathbf{x}_j \mathbb{I} \left\{ \mathbf{w}^\top \mathbf{x}_i \geq 0, \mathbf{w}^\top \mathbf{x}_j \geq 0 \right\} \right] = \mathbf{x}_i^\top \mathbf{x}_j \frac{\pi - \arccos(\mathbf{x}_i^\top \mathbf{x}_j)}{2\pi}. \tag{10}$$

This matrix coincides with neural tangent kernel [Jacot et al., 2018] for ReLU activation function. As shown by Du et al. [2018b], this matrix is positive definite and we define its smallest eigenvalue $\lambda_0 \triangleq \lambda_{\min}(\mathbf{G}^\infty) > 0$. In the same way, we can define its finite version $\mathbf{G}(t) = \mathbf{J}(t)\mathbf{J}(t)^\top$ with $(i, j)$-entry $\mathbf{G}_{ij}(t) = \frac{1}{m}\mathbf{x}_i^\top \mathbf{x}_j \sum_{r \in [m]} \mathbb{I} \left\{ \mathbf{w}_r(t)^\top \mathbf{x}_i \geq 0, \mathbf{w}_r(t)^\top \mathbf{x}_j \geq 0 \right\}$.

### 4.3 Exact Natural Gradient Descent

In this subsection, we present our result for this setting. The main difficulty is to show that Conditions 1 and 2 hold. Here we state our main result.

**Theorem 3** (Natural Gradient Descent for overparameterized Networks). *Under Assumption 1, if we i.i.d initialize $\mathbf{w}_r \sim \mathcal{N}(\mathbf{0}, \nu^2 \mathbf{I})$, $a_r \sim \mathrm{unif}[\{-1, +1\}]$ for $r \in [m]$, we set the number of hidden*

nodes $m = \Omega\left(\frac{n^4}{\nu^2 \lambda_0^4 \delta^3}\right)$, and the step size $\eta = \mathcal{O}(1)$, then with probability at least $1 - \delta$ over the random initialization we have for $k = 0, 1, 2, ...$

$$\|\mathbf{u}(k) - \mathbf{y}\|_2^2 \leq (1 - \eta)^k \|\mathbf{u}(0) - \mathbf{y}\|_2^2. \tag{11}$$

Even though the objective is non-convex and non-smooth, natural gradient descent with a constant step size enjoys a linear convergence rate. For large enough $m$, we show that the learning rate can be chosen up to 1, so NGD can provably converge within $\mathcal{O}(1)$ steps. Compared to analogous bounds for gradient descent [Du et al., 2018a, Oymak and Soltanolkotabi, 2019, Wu et al., 2019], we improve the maximum allowable learning rate from $\mathcal{O}(1/n)$ to $\mathcal{O}(1)$ and also get rid of the dependency on $\lambda_0$. Overall, NGD (Theorem 3) gives an $\mathcal{O}(\lambda_0/n)$ improvement over gradient descent.

Our strategy to prove this result will be to show that for the given choice of random initialization, Condition 1 and 2 hold with high probability. For proving Condition 1 hold, we used matrix concentration inequalities. For Condition 2, we show that $\|\mathbf{J} - \mathbf{J}(0)\|_2 = \mathcal{O}\left(m^{-1/6}\right)$, which implies the Jacobian is stable for wide networks. For detailed proof, we refer the reader to the Appendix D.1.

### 4.4 Approximate Natural Gradient Descent with K-FAC

Exact natural gradient descent is quite expensive in terms of computation or memory. In training deep neural networks, K-FAC [Martens and Grosse, 2015] has been a powerful optimizer for leveraging curvature information while retaining tractable computation. The K-FAC update rule for the two-layer ReLU network is given by

$$\mathbf{w}(k+1) = \mathbf{w}(k) - \eta \underbrace{\left[(\mathbf{X}^\top \mathbf{X})^{-1} \otimes (\mathbf{S}(k)^\top \mathbf{S}(k))^{-1}\right]}_{\mathbf{F}_{\mathrm{K-FAC}}^{-1}} \mathbf{J}(k)^\top (\mathbf{u}(k) - \mathbf{y}). \tag{12}$$

where $\mathbf{X} \in \mathbb{R}^{n \times d}$ denotes the matrix formed from the $n$ input vectors (i.e. $\mathbf{X} = [\mathbf{x}_1, ..., \mathbf{x}_n]^\top$), and $\mathbf{S} = [\phi'(\mathbf{X}\mathbf{w}_1), ..., \phi'(\mathbf{X}\mathbf{w}_m)] \in \mathbb{R}^{n \times m}$ is the matrix of pre-activation derivatives. Under the same argument as the Gram matrix $\mathbf{G}^\infty$, we get that $\mathbf{S}^\infty \mathbf{S}^{\infty\top}$ is strictly positive definite with smallest eigenvalue $\lambda_{\mathbf{S}}$ (see Appendix D.3 for detailed proof).

We show that for sufficiently wide networks, K-FAC does converge linearly to a global minimizer. We further show, with a particular transformation on the input data, K-FAC does match the optimization performance of exact natural gradient for two-layer ReLU networks. Here we state the main result.

**Theorem 4** (K-FAC). *Under the same assumptions as in Theorem 3, plus the additional assumption that* $\mathrm{rank}(\mathbf{X}) = d$, *if we set the number of hidden units* $m = \mathcal{O}\left(\frac{n^4}{\nu^2 \lambda_{\mathbf{S}}^4 \kappa_{\mathbf{X}^\top \mathbf{X}}^4 \delta^3}\right)$ *and step size* $\eta = \mathcal{O}\left(\lambda_{\min}\left(\mathbf{X}^\top \mathbf{X}\right)\right)$, *then with probability at least* $1 - \delta$ *over the random initialization, we have for* $k = 0, 1, 2, ...$

$$\|\mathbf{u}(k) - \mathbf{y}\|_2^2 \leq \left(1 - \frac{\eta}{\lambda_{\max}(\mathbf{X}^\top \mathbf{X})}\right)^k \|\mathbf{u}(0) - \mathbf{y}\|_2^2. \tag{13}$$

The key step in proving Theorem 4 is to show

$$\mathbf{u}(k+1) - \mathbf{u}(k) \approx \left[\left(\mathbf{X}(\mathbf{X}^\top \mathbf{X})^{-1}\mathbf{X}^\top\right) \odot \mathbf{I}\right](\mathbf{y} - \mathbf{u}(k)). \tag{14}$$

**Remark 3.** *The convergence rate of K-FAC is captured by the condition number of the matrix* $\mathbf{X}^\top \mathbf{X}$, *as opposed to gradient descent [Du et al., 2018b, Oymak and Soltanolkotabi, 2019], for which the convergence rate is determined by the condition number of the Gram matrix* $\mathbf{G}$.

**Remark 4.** *The dependence of the convergence rate on* $\kappa_{\mathbf{X}^\top \mathbf{X}}$ *in Theorem 4 may seem paradoxical, as K-FAC is invariant to invertible linear transformations of the data (including those that would change* $\kappa_{\mathbf{X}^\top \mathbf{X}}$*). But we note that said transformations would also make the norms of the input vectors non-uniform, thus violating Assumption 1 in a way that isn't repairable. Interestingly, there exists an invertible linear transformation which, if applied to the input vectors and followed by normalization, produces vectors that simultaneously satisfy Assumption 1 and the condition* $\kappa_{\mathbf{X}^\top \mathbf{X}} = 1$ *(thus improving the bound in Theorem 4 substantially). See Appendix A for details. Notably, K-FAC is not invariant to such pre-processing, as the normalization step is a nonlinear operation.*

To quantify the degree of overparameterization (which is a function of the network width $m$) required to achieve global convergence under our analysis, we must estimate $\lambda_{\mathbf{S}}$. To this end, we observe that $\mathbf{G} = \mathbf{X}\mathbf{X}^\top \odot \mathbf{S}\mathbf{S}^\top$, and then apply the following lemma:

**Lemma 1.** *[Schur [1911]] For two positive definite matrices $\mathbf{A}$ and $\mathbf{B}$, we have*

$$\lambda_{\max}\left(\mathbf{A} \odot \mathbf{B}\right) \leq \max_i \mathbf{A}_{ii} \lambda_{\max}(\mathbf{B})$$
$$\lambda_{\min}\left(\mathbf{A} \odot \mathbf{B}\right) \geq \min_i \mathbf{A}_{ii} \lambda_{\min}(\mathbf{B}) \tag{15}$$

The diagonal entries of $\mathbf{X}\mathbf{X}^\top$ are all 1 since the inputs are normalized. Therefore, we have $\lambda_0 \geq \lambda_{\mathbf{S}}$ according to Lemma 1, and hence K-FAC requires a slightly higher degree of overparameterization than exact NGD under our analysis.

### 4.5 Bounding $\lambda_0$

As pointed out by Allen-Zhu et al. [2018], it is unclear if $1/\lambda_0$ is small or even polynomial. Here, we bound $\lambda$ using matrix concentration inequalities and harmonic analysis. To leverage harmonic analysis, we have to assume the data $\mathbf{x}_i$ are drawn i.i.d. from the unit sphere[3].

**Theorem 5.** *Under this assumption on the training data, with probability $1 - n\exp(-n^\beta/4)$,*

$$\lambda_0 \triangleq \lambda_{\min}(\mathbf{G}^\infty) \geq n^\beta/2, \text{ where } \beta \in (0, 0.5) \tag{16}$$

Basically, Theorem 5 says that the Gram matrix $\mathbf{G}^\infty$ should have high chance of having large smallest eigenvalue if the training data are uniformly distributed. Intuitively, we would expect the smallest eigenvalue to be very small if all $\mathbf{x}_i$ are similar to each other. Therefore, some notion of diversity of the training inputs is needed. We conjecture that the smallest eigenvalue would still be large if the data are $\delta$-separable (i.e., $\|\mathbf{x}_i - \mathbf{x}_j\|_2 \geq \delta$ for any pair $i, j \in [n]$), an assumption adopted by Li and Liang [2018], Allen-Zhu et al. [2018], Zou et al. [2018].

## 5 Generalization analysis

It is often speculated that NGD or other preconditioned gradient descent methods (e.g., Adam) perform worse than gradient descent in terms of generalization [Wilson et al., 2017]. In this section, we show that NGD achieves the same generalization bounds which have been proved for GD, at least for two-layer ReLU networks.

Consider a loss function $\ell : \mathbb{R} \times \mathbb{R} \to \mathbb{R}$. The expected risk over the data distribution $\mathcal{D}$ and the empirical risk over a training set $\mathcal{S} = \{(\mathbf{x}_i, y_i)\}_{i=1}^n$ are defined as

$$\mathcal{L}_{\mathcal{D}}(f) = \mathbb{E}_{(\mathbf{x},y)\sim\mathcal{D}}\left[\ell(f(\mathbf{x}), y)\right] \text{ and } \mathcal{L}_{\mathcal{S}}(f) = \frac{1}{n}\sum_{i=1}^n \ell(f(\mathbf{x}_i), y_i) \tag{17}$$

It has been shown [Neyshabur et al., 2019] that the Redemacher complexity [Bartlett and Mendelson, 2002] for two-layer ReLU networks depends on $\|\mathbf{w} - \mathbf{w}(0)\|_2$. By the standard Rademacher complexity generalization bound, we have the following bound (see Appendix E.1 for proof):

**Theorem 6.** *Given a target error parameter $\epsilon > 0$ and failure probability $\delta \in (0, 1)$. Suppose $\nu = \mathcal{O}\left(\epsilon\sqrt{\lambda_0\delta}\right)$ and $m \geq \nu^{-2}\text{poly}\left(n, \lambda_0^{-1}, \delta^{-1}, \epsilon^{-1}\right)$. For any 1-Lipschitz loss function, with probability at least $1 - \delta$ over random initialization and training samples, the two-layer neural network $f(\mathbf{w}, \mathbf{a})$ trained by NGD for $k \geq \Omega\left(\frac{1}{\eta}\log\frac{1}{\epsilon\delta}\right)$ iterations has expected loss $\mathcal{L}_{\mathcal{D}}(f(\mathbf{w}, \mathbf{a})) = \mathbb{E}_{(\mathbf{x},y)\sim\mathcal{D}}\left[\ell(f(\mathbf{w}, \mathbf{a}, \mathbf{x}), y)\right]$ bounded as:*

$$\mathcal{L}_{\mathcal{D}}(f(\mathbf{w}, \mathbf{a})) \leq \sqrt{\frac{2\mathbf{y}^\top(\mathbf{G}^\infty)^{-1}\mathbf{y}}{n}} + 3\sqrt{\frac{\log(6/\delta)}{2n}} + \epsilon \tag{18}$$

which matches the bound for gradient descent in Arora et al. [2019b]. For detailed proof, we refer the reader to the Appendix E.1.

# 6 Conclusion

We've analyzed for the first time the rate of convergence to a global optimum for (both exact and approximate) natural gradient descent on nonlinear neural networks. Particularly, we identified two conditions which guarantee the global convergence, i.e., the Jacobian matrix with respect to the parameters has full row rank and stable for perturbations around the initialization. Based on these insights, we improved the convergence rate of gradient descent by a factor of $\mathcal{O}(\lambda_0/n)$ on two-layer ReLU networks by using natural gradient descent. Beyond that, we also showed that the improved convergence rates don't come at the expense of worse generalization.

## Acknowledgements

We thank Jeffrey Z. HaoChen, Shengyang Sun and Mufan Li for helpful discussion. RG acknowledges support from the CIFAR Canadian AI Chairs program and the Ontario MRIS Early Researcher Award.

## Footnotes

[1]It is easy to extend to multi-output networks and other loss functions, here we focus on single-output and quadratic just for notational simplicity.

[2]We fix the second layer just for simplicity. Based on the same analysis, one can also prove global convergence for jointly training both layers.

[3]This assumption is not too stringent since the inputs are already normalized. Moreover, we can relax the assumption of unit sphere input to separable input, which is used in Li and Liang [2018], Allen-Zhu et al. [2018], Zou et al. [2018]. See Oymak and Soltanolkotabi [2019] (Theorem I.1) for more details.

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
