[Supplementary Material]

# A  The Forster Transform

In a breakthrough paper in the area of communication complexity, Forster [2002] used the existence of a certain kind of dataset transformation as the key technical tool in the proof of his main result. The Theorem which establishes the existence of this transformation is paraphrased below.

**Theorem 7** (Forster [2002], Theorem 4.1). *Suppose $\mathbf{X} \in \mathbb{R}^{n \times d}$ is a matrix such that all subsets of size at most $d$ of its rows are linearly independent. Then there exists an invertible matrix $\mathbf{A} \in \mathbb{R}^{d \times d}$ such that if we post-multiply $\mathbf{X}$ by $\mathbf{A}$ (i.e. apply $\mathbf{A}$ to each row), and then normalize each row by its 2-norm, the resulting matrix $\mathbf{Z} \in \mathbb{R}^{n \times d}$ satisfies $\mathbf{Z}^\top \mathbf{Z} = \frac{n}{d} \mathbf{I}_d$.*

**Remark 5.** *Note that the technical condition about linear independence can be easily be made to hold for an arbitrary $\mathbf{X}$ by adding an infinitesimal random perturbation, assuming it doesn't hold to begin with.*

This result basically says that for any set of vectors, there is a linear transformation of said vectors which makes their normalized versions (given by the rows of $\mathbf{Z}$) satisfy $\mathbf{Z}^\top \mathbf{Z} = \frac{n}{d} \mathbf{I}_d$. So by combining this linear transformation with normalization we produce a set of vectors that simultaneously satisfy Assumption 1, while also satisfying $\kappa_{\mathbf{Z}^\top \mathbf{Z}} = 1$.

Forster's proof of Theorem 7 can be interpreted as defining a transformation function on $\mathbf{Z}$ (initialized at $\mathbf{X}$), and showing that it has a fixed point with the required properties. One can derive an algorithm from this by repeatedly applying the transformation to $\mathbf{Z}$, which consists of "whitening" followed by normalization, until $\mathbf{Z}^\top \mathbf{Z}$ is sufficiently close to $\frac{n}{d} \mathbf{I}_d$. The $\mathbf{A}$ matrix is then simply the product of the "whitening" transformation matrices, up to a scalar constant. While no explicit finite-time convergence guarantees are given for this algorithm by Forster [2002], we have implemented it and verified that it does indeed converge at a reasonable rate. The algorithm is outlined below.

---

**Algorithm 1** Forster Transform

---

1: **INPUT:** Matrix $\mathbf{X} \in \mathbb{R}^{n \times d}$ satisfying the hypotheses of Theorem 7
2: $\mathbf{Z} \leftarrow \mathbf{X}$
3: $\mathbf{A} \leftarrow \mathbf{I}_d$
4: **while** error tolerance exceeded **do**
5: $\quad \mathbf{T} \leftarrow (\mathbf{Z}^\top \mathbf{Z})^{-\frac{1}{2}}$
6: $\quad \mathbf{Z} \leftarrow \mathbf{Z}\mathbf{T}$
7: $\quad \mathbf{Z} \leftarrow \text{normalize-rows}(\mathbf{Z})$
8: $\quad \mathbf{A} \leftarrow \mathbf{A}\mathbf{T}$
9: $\quad \mathbf{A} \leftarrow \frac{1}{[\mathbf{A}]_{1,1}} \mathbf{A}$
10: **end while**
11: **OUTPUT:** $\mathbf{A} \in \mathbb{R}^{d \times d}$ with the properties stated in Theorem 7 (up to an error tolerance)

---

# B  Proof of Theorem 1

We prove the result in two steps: we first provide a convergence analysis for natural gradient flow, i.e., natural gradient descent with infinitesimal step size, and then take into account the error introduced by discretization and show global convergence for natural gradient descent.

To guarantee global convergence for natural gradient flow, we only need to show that the Gram matrix is positive definite throughout the training. Intuitively, for successfully finding global minima, the network must satisfy the following condition, i.e., the gradient with respect to the parameters $\nabla_{\boldsymbol{\theta}} \mathcal{L}(\boldsymbol{\theta})$ is zero only if the gradient in the output space $\nabla_{\mathbf{u}} \mathcal{L}(\boldsymbol{\theta}) = \mathbf{0}$ is zero. It suffices to show that the Gram matrix is positive definite, or equivalently, the Jacobian matrix is full row rank.

By Condition 1 and Condition 2, we immediately obtain the following lemma that if the parameters stay close to the initialization, then the Gram matrix is positive definite throughout the training.

**Lemma 2.** *If $\|\boldsymbol{\theta} - \boldsymbol{\theta}(0)\|_2 \leq \frac{3\|\mathbf{y} - \mathbf{u}(0)\|_2}{\sqrt{\lambda_{\min}(\mathbf{G}(0))}}$, then we have $\lambda_{\min}(\mathbf{G}) \geq \frac{4}{9}\lambda_{\min}(\mathbf{G}(0))$.*

*Proof of Lemma 2.* Based on the inequality that $\sigma_{\min}(\mathbf{A} + \mathbf{B}) \geq \sigma_{\min}(\mathbf{A}) - \sigma_{\max}(\mathbf{B})$ where $\sigma$ denotes singular value, we have

$$\sigma_{\min}(\mathbf{J}) \geq \sigma_{\min}(\mathbf{J}(0)) - \|\mathbf{J} - \mathbf{J}(0)\|_2 \tag{19}$$

By Condition 2, we have $\|\mathbf{J} - \mathbf{J}(0)\|_2 \leq \frac{1}{3}\sqrt{\lambda_{\min}(\mathbf{G}(0))}$, thus we get $\sigma_{\min}(\mathbf{J}) \geq \frac{2}{3}\sqrt{\lambda_{\min}(\mathbf{G}(0))}$ which completes the proof. $\qquad\square$

With the assumption that $\|\boldsymbol{\theta} - \boldsymbol{\theta}(0)\|_2 \leq \frac{3\|\mathbf{y}-\mathbf{u}(0)\|_2}{\sqrt{\lambda_{\min}(\mathbf{G}(0))}}$ throughout the training, we are now ready to prove global convergence for natural gradient flow. Recall the dynamics of natural gradient flow in weight space,

$$\frac{d}{dt}\boldsymbol{\theta}(t) = \frac{1}{n}\mathbf{F}(t)^{\dagger}\mathbf{J}(t)^{\top}(\mathbf{y} - \mathbf{u}(t)) \tag{20}$$

Accordingly, we can calculate the dynamics of the network predictions.

$$\begin{aligned}
\frac{d}{dt}\mathbf{u}(t) &= \frac{1}{n}\mathbf{J}(t)\mathbf{F}(t)^{\dagger}\mathbf{J}(t)^{\top}(\mathbf{y} - \mathbf{u}(t)) \\
&= \mathbf{J}(t)\mathbf{J}(t)^{\top}\mathbf{G}(t)^{-1}\mathbf{G}(t)^{-1}\mathbf{J}(t)\mathbf{J}(t)^{\top}(\mathbf{y} - \mathbf{u}(t))
\end{aligned} \tag{21}$$

Since the Gram matrix $\mathbf{G}(t)$ is positive definite, its inverse does exist. Therefore, we have

$$\frac{d}{dt}\mathbf{u}(t) = \mathbf{y} - \mathbf{u}(t) \tag{22}$$

By the chain rule, we get the dynamics of the loss in the following form:

$$\frac{d}{dt}\|\mathbf{y} - \mathbf{u}(t)\|_2^2 = -2(\mathbf{y} - \mathbf{u}(t))^{\top}(\mathbf{y} - \mathbf{u}(t)) \tag{23}$$

By integrating eqn. (23), we find that $\|\mathbf{y} - \mathbf{u}(t)\|_2^2 = \exp(-2t)\|\mathbf{y} - \mathbf{u}(0)\|_2^2$.

That completes the continuous time analysis, under the assumption that the parameters stay close to the initialization. The discrete case follows similarly, except that we need to account for the discretization error. Analogously to eqn. (21), we calculate the difference of predictions between two consecutive iterations.

$$\begin{aligned}
\mathbf{u}(k+1) - \mathbf{u}(k) &= \mathbf{u}\left(\boldsymbol{\theta}(k) - \eta\mathbf{J}(k)^{\top}\mathbf{G}(k)^{-1}(\mathbf{u}(k) - \mathbf{y})\right) - \mathbf{u}(\boldsymbol{\theta}(k)) \\
&= -\int_{s=0}^{1}\left\langle \frac{\partial\mathbf{u}(\boldsymbol{\theta}(s))}{\partial\boldsymbol{\theta}^{\top}}, \eta\mathbf{J}(k)^{\top}\mathbf{G}(k)^{-1}(\mathbf{u}(k) - \mathbf{y})\right\rangle ds \\
&= \underbrace{-\int_{s=0}^{1}\left\langle \frac{\partial\mathbf{u}(\boldsymbol{\theta}(k))}{\partial\boldsymbol{\theta}^{\top}}, \eta\mathbf{J}(k)^{\top}\mathbf{G}(k)^{-1}(\mathbf{u}(k) - \mathbf{y})\right\rangle ds}_{\eta(\mathbf{y}-\mathbf{u}(k))} \\
&\quad + \underbrace{\int_{s=0}^{1}\left\langle \frac{\partial\mathbf{u}(\boldsymbol{\theta}(k))}{\partial\mathbf{w}^{\top}} - \frac{\partial\mathbf{u}(\boldsymbol{\theta}(s))}{\partial\boldsymbol{\theta}^{\top}}, \eta\mathbf{J}(k)^{\top}\mathbf{G}(k)^{-1}(\mathbf{u}(k) - \mathbf{y})\right\rangle ds}_{\textcircled{1}},
\end{aligned} \tag{24}$$

where we have defined $\boldsymbol{\theta}(s) = s\boldsymbol{\theta}(k+1) + (1-s)\boldsymbol{\theta}(k) = \boldsymbol{\theta}(k) - s\eta\mathbf{J}(k)^{\top}\mathbf{G}(k)^{-1}(\mathbf{u}(k) - \mathbf{y})$.

Next we bound the norm of the second term ($\textcircled{1}$) in the RHS of eqn. (24). Using Condition 2 and Lemma 2 we have that

$$\begin{aligned}
\left\|\textcircled{1}\right\|_2 &\leq \eta\left\|\int_{s=0}^{1}\mathbf{J}(\boldsymbol{\theta}(s)) - \mathbf{J}(\boldsymbol{\theta}(k))\,ds\right\|_2\left\|\mathbf{J}(k)^{\top}\mathbf{G}(k)^{-1}(\mathbf{u}(k) - \mathbf{y})\right\|_2 \\
&\leq \eta\frac{2C}{3}\sqrt{\lambda_{\min}(\mathbf{G}(0))}\frac{1}{\sqrt{\lambda_{\min}(\mathbf{G}(k))}}\|\mathbf{u}(k) - \mathbf{y}\|_2 \\
&\leq \eta\frac{2C}{3}\sqrt{\lambda_{\min}(\mathbf{G}(0))}\frac{3}{2\sqrt{\lambda_{\min}(\mathbf{G}(0))}}\|\mathbf{u}(k) - \mathbf{y}\|_2 \\
&= \eta C\|\mathbf{u}(k) - \mathbf{y}\|_2.
\end{aligned} \tag{25}$$

In the first inequality, we used the fact (based on Condition 2) that

$$\left\| \int_{s=0}^{1} \mathbf{J}(\boldsymbol{\theta}(s)) - \mathbf{J}(\boldsymbol{\theta}(k))\, ds \right\|_2 \leq \|\mathbf{J}(\theta(k)) - \mathbf{J}(\theta(0))\|_2 + \|\mathbf{J}(\theta(k+1)) - \mathbf{J}(\theta(0))\|_2$$

$$\leq \frac{2C}{3}\sqrt{\lambda_{\min}(\mathbf{G}(0))} \tag{26}$$

Lastly, we have

$$\|\mathbf{y} - \mathbf{u}(k+1)\|_2^2 = \|\mathbf{y} - \mathbf{u}(k) - (\mathbf{u}(k+1) - \mathbf{u}(k))\|_2^2$$
$$= \|\mathbf{y} - \mathbf{u}(k)\|_2^2 - 2(\mathbf{y} - \mathbf{u}(k))^\top (\mathbf{u}(k+1) - \mathbf{u}(k)) + \|\mathbf{u}(k+1) - \mathbf{u}(k)\|_2^2$$
$$\leq \left(1 - 2\eta + 2\eta C + \eta^2 (1 + C)^2\right) \|\mathbf{y} - \mathbf{u}(k)\|_2^2$$
$$\leq (1 - \eta) \|\mathbf{y} - \mathbf{u}(k)\|_2^2. \tag{27}$$

In the last inequality of eqn. (27), we use the assumption that $\eta \leq \frac{1-2C}{(1+C)^2}$.

So far, we have assumed the parameters fall within a certain radius around the initialization. We now justify this assumption.

**Lemma 3.** *If Conditions 1 and 2 hold, then as long as $\lambda_{\min}(\mathbf{G}(k)) \geq \frac{4}{9}\lambda_{\min}(\mathbf{G}(0))$, we have*

$$\|\boldsymbol{\theta}(k+1) - \boldsymbol{\theta}(0)\|_2 \leq \frac{3\|\mathbf{y} - \mathbf{u}(0)\|_2}{\sqrt{\lambda_{\min}(\mathbf{G}(0))}}. \tag{28}$$

*Proof of Lemma 3.* We use the norm of each update to bound the distance of the parameters to the initialization.

$$\|\boldsymbol{\theta}(k+1) - \boldsymbol{\theta}(0)\|_2 \leq \eta \sum_{s=0}^{k} \|\mathbf{J}(s)^\top \mathbf{G}(s)^\top (\mathbf{y} - \mathbf{u}(s))\|_2$$
$$\leq \eta \sum_{s=0}^{k} \frac{\|\mathbf{y} - \mathbf{u}(s)\|_2}{\sqrt{\lambda_{\min}(\mathbf{G}(s))}}$$
$$\leq \eta \sum_{s=0}^{k} \frac{(1-\eta)^{s/2}\|\mathbf{y} - \mathbf{u}(0)\|_2}{\sqrt{\frac{4}{9}\lambda_{\min}(\mathbf{G}(0))}}$$
$$\leq \frac{3\|\mathbf{y} - \mathbf{u}(0)\|_2}{\sqrt{\lambda_{\min}(\mathbf{G}(0))}}. \tag{29}$$

This completes the proof. □

At first glance, the proofs in Lemma 2 and 3 seem to be circular. Here, we prove that their assumptions continue to be jointly satisfied.

**Lemma 4.** *Assuming Conditions 1 and 2, we have (1) $\|\boldsymbol{\theta}(k) - \boldsymbol{\theta}(0)\|_2 \leq \frac{3\|\mathbf{y}-\mathbf{u}(0)\|_2}{\sqrt{\lambda_{\min}(\mathbf{G}(0))}}$ and (2) $\lambda_{\min}(\mathbf{G}(k)) \geq \frac{4}{9}\lambda_{\min}(\mathbf{G}(0))$ throughout the training.*

*Proof.* We prove the lemma by contradiction. Suppose the conclusion does not hold for all iterations. Let's say (1) holds at iteration $k = 0, ..., k_0$ but not iteration $k_0 + 1$. Then we know, there must exist $0 < k' \leq k_0$ such that from $\lambda_{\min}(\mathbf{G}(k')) < \frac{4}{9}\lambda_{\min}(\mathbf{G}(0))$, otherwise we can show that (1) holds at iteration $k_0 + 1$ as well by Lemma 3. However, by Lemma 2, we know that $\lambda_{\min}(\mathbf{G}(k')) \geq \frac{4}{9}\lambda_{\min}(\mathbf{G}(0))$ since (1) holds for $k = 0, ..., k_0$, contradiction. □

Notably, Lemma 4 shows that $\|\boldsymbol{\theta}(k) - \boldsymbol{\theta}(0)\|_2 \leq \frac{3\|\mathbf{y}-\mathbf{u}(0)\|_2}{\sqrt{\lambda_{\min}(\mathbf{G}(0))}}$ and $\lambda_{\min}(\mathbf{G}(k)) \geq \frac{4}{9}\lambda_{\min}(\mathbf{G}(0))$ throughout the training if Conditions 1 and 2 hold. This completes the proof of our main result.

## C   Proof of Theorem 2

Here, we prove Theorem 2 by induction. Our inductive hypothesis is the following condition.

**Condition 4.** *At the $k$-th iteration, we have* $\|\mathbf{y} - \mathbf{u}(k+1)\|_2^2 \leq \left(1 - \frac{2\eta\mu L}{\mu+L}\right) \|\mathbf{y} - \mathbf{u}(k)\|_2^2$.

We first use the norm of the gradient to bound the distance of the weights. Here we slightly abuse notation, $\mathcal{L}(\mathbf{u}) = \sum_{i=1}^{n} \ell(u_i, y_i)$.

$$
\begin{aligned}
\|\boldsymbol{\theta}(k+1) - \boldsymbol{\theta}(0)\|_2 &\leq \eta \sum_{s=0}^{k} \|\mathbf{J}(k)^\top \mathbf{G}(k)^{-1} \nabla_{\mathbf{u}} \mathcal{L}(\mathbf{u}(k))\|_2 \\
&\leq \eta L \sum_{s=0}^{k} \|\mathbf{J}(k)^\top \mathbf{G}(k)^{-1}\|_2 \|\mathbf{y} - \mathbf{u}(k)\|_2 \\
&\leq \eta L \sum_{s=0}^{\infty} \left(1 - \frac{2\eta\mu L}{\mu+L}\right)^{s/2} \frac{\|\mathbf{y} - \mathbf{u}(0)\|_2}{\sqrt{\frac{4}{9}\lambda_{\min}(\mathbf{G}(0))}} \\
&= \frac{3(1+\kappa)\|\mathbf{y} - \mathbf{u}(0)\|_2}{2\sqrt{\lambda_{\min}(\mathbf{G}(0))}}
\end{aligned}
\tag{30}
$$

where $\kappa = \frac{L}{\mu}$. The second inequality is based on the $L$-Lipschitz gradient assumption[4] and the fact that $\nabla_{\mathbf{u}} \mathcal{L}(\mathbf{y}) = 0$. Also, we have

$$
\mathbf{u}(k+1) - \mathbf{u}(k) = \eta \nabla_{\mathbf{u}} \mathcal{L}(\mathbf{u}(k)) + \eta \mathbf{P}(k) \nabla_{\mathbf{u}} \mathcal{L}(\mathbf{u}(k))
\tag{31}
$$

In analogy to eqn. (24), $\mathbf{P}(k) = \int_{s=0}^{1} \left(\mathbf{J}(k) - \mathbf{J}(\boldsymbol{\theta}(s))\right) \mathbf{J}(k)^\top \mathbf{G}(k)^{-1} ds$. Next, we introduce a well-known Lemma for $\mu$-strongly convex and $L$-Lipschitz gradient loss.

**Lemma 5** (Co-coercivity for $\mu$-strongly convex loss). *If the loss function is $\mu$-strongly convex with $L$-Lipschitz gradient, for any $\mathbf{u}, \mathbf{y} \in \mathbb{R}^n$, the following inequality holds.*

$$
\left(\nabla \mathcal{L}(\mathbf{u}) - \nabla \mathcal{L}(\mathbf{y})\right)^\top (\mathbf{u} - \mathbf{y}) \geq \frac{\mu L}{\mu+L} \|\mathbf{u} - \mathbf{y}\|_2^2 + \frac{1}{\mu+L} \|\nabla \mathcal{L}(\mathbf{u}) - \nabla \mathcal{L}(\mathbf{y})\|_2^2
\tag{32}
$$

Now, we are ready to bound $\|\mathbf{y} - \mathbf{u}(k+1)\|_2^2$:

$$
\begin{aligned}
\|\mathbf{u}(k+1) - \mathbf{y}\|_2^2 &= \|\mathbf{u}(k) - \eta(\mathbf{I} + \mathbf{P}(k))\nabla_{\mathbf{u}}\mathcal{L}(\mathbf{u}(k)) - \mathbf{y}\|_2^2 \\
&\leq \|\mathbf{u}(k) - \mathbf{y}\|_2^2 - 2\eta \nabla_{\mathbf{u}}\mathcal{L}(\mathbf{u}(k))^\top (\mathbf{u}(k) - \mathbf{y}) \\
&\quad + 2\eta \|\mathbf{P}\|_2 \|\nabla_{\mathbf{u}}\mathcal{L}(\mathbf{u}(k))\|_2 \|\mathbf{u}(k) - \mathbf{y}\|_2 + \eta^2 (1 + \|\mathbf{P}(k)\|_2)^2 \|\nabla_{\mathbf{u}}\mathcal{L}(\mathbf{u}(k))\|_2^2 \\
&\leq \|\mathbf{u}(k) - \mathbf{y}\|_2^2 - 2\eta \frac{\mu L}{\mu+L} \|\mathbf{u}(k) - \mathbf{y}\|_2^2 - \frac{2\eta}{\mu+L} \|\nabla_{\mathbf{u}}\mathcal{L}(\mathbf{u}(k))\|_2^2 \\
&\quad + 2\eta \|\mathbf{P}\|_2 \|\nabla_{\mathbf{u}}\mathcal{L}(\mathbf{u}(k))\|_2 \|\mathbf{u}(k) - \mathbf{y}\|_2 + \eta^2 (1 + \|\mathbf{P}(k)\|_2)^2 \|\nabla_{\mathbf{u}}\mathcal{L}(\mathbf{u}(k))\|_2^2 \\
&\leq \left(1 - \frac{2\eta\mu L}{\mu+L}\right) \|\mathbf{u}(k) - \mathbf{y}\|_2^2 + \eta^2 (1 + C)^2 \|\nabla_{\mathbf{u}}\mathcal{L}(\mathbf{u}(k))\|_2^2 \\
&\quad + 2\eta C \|\nabla_{\mathbf{u}}\mathcal{L}(\mathbf{u}(k))\|_2 \|\mathbf{u}(k) - \mathbf{y}\|_2 - \frac{2\eta}{\mu+L} \|\nabla_{\mathbf{u}}\mathcal{L}(\mathbf{u}(k))\|_2^2 \\
&\leq \left(1 - \frac{2\eta\mu L}{\mu+L}\right) \|\mathbf{u}(k) - \mathbf{y}\|_2^2
\end{aligned}
\tag{33}
$$

For the second inequality we used Lemma 5 and the fact that $\nabla_{\mathbf{u}}\mathcal{L}(\mathbf{y}) = \mathbf{0}$. For the third inequality we used the result of Condition 3 that $\|\mathbf{P}(\mathbf{u}(k))\|_2 \leq C$. For the last inequality we used the fact that the loss is $\mu$-strongly convex and $\eta < \frac{2}{\mu+L} \frac{1 - (1+\kappa)C}{(1+C)^2}$.

*Proof of Lemma 5.* For convex function $f$ with Lipschtiz gradient, we have the following co-coercivity property [Boyd and Vandenberghe, 2004]:

$$\left(\nabla f(\mathbf{x}) - \nabla f(\mathbf{y})\right)^\top \left(\mathbf{x} - \mathbf{y}\right) \geq \frac{1}{L} \left\|\nabla f(\mathbf{x}) - \nabla f(\mathbf{y})\right\|_2^2 \tag{34}$$

Then, for $\mu$-strongly convex loss, we can define $g(\mathbf{x}) = f(\mathbf{x}) - \frac{\mu}{2}\|\mathbf{x}\|_2^2$, which is a convex function with $L - \mu$ Lipschitz gradient. By co-coercivity of $g$, we get

$$\left(\nabla g(\mathbf{x}) - \nabla g(\mathbf{y})\right)^\top \left(\mathbf{x} - \mathbf{y}\right) \geq \frac{1}{L - \mu} \left\|\nabla g(\mathbf{x}) - \nabla g(\mathbf{y})\right\|_2^2 \tag{35}$$

After plugging $\nabla g(\mathbf{x}) = \nabla f(\mathbf{x}) - \mu\mathbf{x}$ and some manipulations, we have

$$\left(\nabla f(\mathbf{x}) - \nabla f(\mathbf{y})\right)^\top \left(\mathbf{x} - \mathbf{y}\right) \geq \frac{\mu L}{\mu + L}\|\mathbf{x} - \mathbf{y}\|_2^2 + \frac{1}{\mu + L}\|\nabla f(\mathbf{x}) - \nabla f(\mathbf{y})\|_2^2 \tag{36}$$

This completes the proof. $\qquad\square$

# D  Proofs for Section 4

## D.1  Proof of Theorem 3

Our strategy to prove this result will be to show that for the given choice of random initialization, Conditions 1 and 2 hold with high probability.

We start with Condition 1, which requires that the inital Gram matrix $\mathbf{G}(0)$ is non-singular, or equivalently that $\mathbf{J}(0)$ has full row rank. Because $\mathbf{G}(0)$ is a sum of random matrices, where the expectation of each random matrix is $\frac{1}{m}\mathbf{G}^\infty$, we can bound its smallest eigenvalue using matrix concentration inequalities. Doing so gives us the following lemma.

**Lemma 6.** *If* $m = \Omega\left(\frac{n\log(n/\delta)}{\lambda_0}\right)$, *then we have with probability at least* $1 - \delta$ *that* $\lambda_{\min}(\mathbf{G}(0)) \geq \frac{3}{4}\lambda_0$ *(which implies that* $\mathbf{J}(0)$ *has full row rank).*

Next we introduce another lemma, which we will use to show that Condition 2 holds with high probability.

**Lemma 7.** *With probability at least* $1 - \delta$*, for all weight vectors* $\mathbf{w}$ *that satisfy* $\|\mathbf{w} - \mathbf{w}(0)\|_2 \leq R$*, we have the following bound:*

$$\|\mathbf{J} - \mathbf{J}(0)\|_2^2 \leq \frac{2nR^{2/3}}{\nu^{2/3}\delta^{2/3}m^{1/3}}. \tag{37}$$

Notably, Lemma 7 says that as long as the weights are close to the random initialization, the corresponding Jacobian matrix also stays close to the Jacobian matrix of inital weights. Therefore, we might expect that if the distance to the initialization is sufficiently small, then Condition 2 would hold with high probability.

By taking $R = \frac{3\|\mathbf{y} - \mathbf{u}(0)\|_2}{\sqrt{\lambda_{\min}(\mathbf{G}(0)}}$ in eqn. (37), we have $\|\mathbf{J} - \mathbf{J}(0)\|_2^2 \leq \frac{96^{1/3}n|\mathbf{y} - \mathbf{u}(0)\|_2^{2/3}}{\nu^{2/3}\lambda_0^{1/3}\delta^{2/3}m^{1/3}}$. Therefore, we know that Condition 2 holds if $m = \Omega\left(\frac{n^3\|\mathbf{y} - \mathbf{u}(0)\|_2^2}{\nu^2\lambda_0^4\delta^2}\right)$. Furthermore, by Assumption 1 we have

$$\mathbb{E}\left[\|\mathbf{y} - \mathbf{u}(0)\|_2^2\right] = \|\mathbf{y}\|_2^2 + 2\mathbf{y}^\top\mathbb{E}\left[\mathbf{u}(0)\right] + \mathbb{E}\left[\|\mathbf{u}(0)\|_2^2\right] = \mathcal{O}\left(n\right). \tag{38}$$

Thus by Markov's inequality, we have probability at least $1 - \delta$, $\|\mathbf{y} - \mathbf{u}(0)\|_2^2 = \mathcal{O}\left(\frac{n}{\delta}\right)$. Thus the condition on $m$ can be written as $m = \Omega\left(\frac{n^4}{\nu^2\lambda_0^4\delta^3}\right)$. We note that the larger $m$ is, the smaller the constant $C$ in Condition 2 is ($C \sim \mathcal{O}\left(m^{-1/6}\right)$). We now finish the proof.

*Proof of Lemma 6.* Notice that $\mathbf{G}(0)$ can be written as the sum of random symmetric matrices:

$$\mathbf{G}(0) = \sum_{r=1}^{m}\mathbf{G}(\mathbf{w}_r) \tag{39}$$

where $\mathbf{G}_{ij}(\mathbf{w}_r) = \frac{1}{m}\mathbf{x}_i^\top \mathbf{x}_j \mathbb{I}\{\mathbf{w}_r^\top \mathbf{x}_i \geq 0, \mathbf{w}_r^\top \mathbf{x}_j \geq 0\}$. Furthermore, $\mathbf{G}(\mathbf{w}_r)$ are positive semi-definite and $\|\mathbf{G}(\mathbf{w}_r)\|_2 \leq \mathrm{tr}\,(\mathbf{G}(\mathbf{w}_r)) = \frac{n}{m}$. Therefore, we can apply the matrix Chernoff bound (e.g., Tropp et al. [2015]), giving us the following bound:

$$\mathbb{P}\left[\lambda_{\min}(\mathbf{G}(0)) \leq (1-\frac{1}{4})\lambda_0\right] \leq n\exp\left(-\frac{1}{4^2}\frac{\lambda_0}{n/m}\right) \tag{40}$$

Letting the RHS of eqn. (40) be $\delta$, we have $m = \mathcal{O}\left(\frac{n}{\lambda_0}\log\frac{n}{\delta}\right)$. $\qquad\square$

*Proof of Lemma 7.* Let $[\mathbf{v}_r]_{k-}$ denote the $k$-th smallest entry of $\{\mathbf{v}_1, ..., \mathbf{v}_m\}$ after sorting its entries in terms of absolute value. We first state a intermediate lemma we prove later.

**Lemma 8.** *Given an integer $k$, suppose*

$$\|\mathbf{w} - \mathbf{w}(0)\|_2 \leq \sqrt{k}\left[\mathbf{w}_r(0)^\top \mathbf{x}_i\right]_{k-} \tag{41}$$

*holds for $i = 1, 2, ..., n$. Then, we have*

$$\|\mathbf{J} - \mathbf{J}(0)\|_2^2 \leq \frac{2nk}{m} \tag{42}$$

By taking $k = \frac{R^{2/3}m^{2/3}}{\nu^{2/3}\delta^{2/3}}$, we have $\|\mathbf{J} - \mathbf{J}(0)\|_2^2 \leq \frac{2nR^{2/3}}{\nu^{2/3}\delta^{2/3}m^{1/3}}$. To complete the proof, all that remains is to prove $R \leq \sqrt{k}\left[\mathbf{w}_r(0)^\top \mathbf{x}_i\right]_{k-}$.

Observe that $\mathbf{w}_r(0)^\top \mathbf{x}_i$ for $r \in [m]$ and $i \in [n]$ all distribute as $\mathcal{N}(0, \nu^2)$, though they depend on each other. We begin by proving that with probability at least $1 - \delta$, at most $k$ of the hidden units, we have $|\mathbf{w}_r(0)^\top \mathbf{x}_i| \leq \frac{k\nu\delta}{m}$. To this aim, we define $\gamma_\alpha$ to be the number for which $\mathbb{P}[|g| \leq \gamma_\alpha] = \alpha$ where $g \sim \mathcal{N}(0, \nu^2)$. By anti-concentration of Gaussian, $\gamma_\alpha$ trivially obeys $\gamma_\alpha \geq \sqrt{\pi/2}\alpha\nu$. Now let $\alpha = \frac{k\delta}{m}$. We note that

$$\mathbb{E}\left[\sum_{r=1}^m \mathbb{I}\{|\mathbf{w}_r(0)^\top \mathbf{x}_i| \leq \gamma_\alpha\}\right] = \sum_{r=1}^m \mathbb{P}\left[|\mathbf{w}_r(0)^\top \mathbf{x}_i| \leq \gamma_\alpha\right] = k\delta \tag{43}$$

By applying Markov's inequality, we obtain

$$\mathbb{P}\left[\sum_{r=1}^m \mathbb{I}\{|\mathbf{w}_r(0)^\top \mathbf{x}_i| \leq \gamma_\alpha\} \geq k\right] \leq \delta \tag{44}$$

Therefore, we have $\sqrt{k}\left[\mathbf{w}_r(0)^\top \mathbf{x}_i\right]_{k-} \geq \frac{k^{3/2}\nu\delta}{m} = R$. $\qquad\square$

*Proof of Lemma 8.* We prove the lemma by contradiction. First, we define the event

$$A_{ir} = \left\{\mathbb{I}\{\mathbf{x}_i^\top \mathbf{w}_r \geq 0\} \neq \mathbb{I}\{\mathbf{x}_i^\top \mathbf{w}_r(0) \geq 0\}\right\} \tag{45}$$

We can then bound the Jacobian perturbation.

$$\|\mathbf{J} - \mathbf{J}(0)\|_2^2 \leq \|\mathbf{J} - \mathbf{J}(0)\|_{\mathbf{F}}^2 \leq \frac{1}{m}\sum_{i=1}^n \sum_{r=1}^m \mathbb{I}\{A_{ir}\} = \frac{2nk}{m} \tag{46}$$

If eqn. (46) does not hold, then we know there must exist an $\mathbf{x}_i$ with $i \in [n]$ such that $\mathbb{I}\{A_{ir}\}$ with $r \in [m]$ has at least $2k$ non-zero entries. Let $\{(a_r, b_r)\}_{r=1}^{2k}$ be entries of $\mathbf{w}_r^\top \mathbf{x}_i$ and $\mathbf{w}_r(0)^\top \mathbf{x}_i$ at these non-zero locations respectively, then we have (by definition, $|b_r| \geq \{\mathbf{w}_r(0)^\top \mathbf{x}_i\}_{k-}$)

$$\begin{aligned}\|\mathbf{w} - \mathbf{w}(0)\|_2^2 &\geq \sum_{r=1}^m |\mathbf{w}_r^\top \mathbf{x}_i - \mathbf{w}_r(0)^\top \mathbf{x}_i|^2 \\ &\geq \sum_{r=1}^{2k} |a_r - b_r|^2 \geq \sum_{r=1}^{2k} |b_r|^2 \\ &\geq k\left\{\mathbf{w}_r(0)^\top \mathbf{x}_i\right\}_{k-}^2\end{aligned} \tag{47}$$

The second inequality, we used the fact that $\mathrm{sign}(a_r) \neq \mathrm{sign}(b_r)$. $\qquad\square$

## D.2 Proof of Theorem 4

Based on the update in the weight space, we can get the update in output space accordingly.

$$
\begin{aligned}
\mathbf{u}(k+1) - \mathbf{u}(k) &= \mathbf{u}\left(\mathbf{w}(k) - \eta\mathbf{F}_{\mathrm{K-FAC}}^{-1}(k)\mathbf{J}(k)^{\top}(\mathbf{u}(k) - \mathbf{y})\right) - \mathbf{u}(\mathbf{w}(k)) \\
&= -\int_{s=0}^{1}\left\langle \frac{\partial\mathbf{u}\left(\mathbf{w}(s)\right)}{\partial\mathbf{w}^{\top}}, \eta\mathbf{F}_{\mathrm{K-FAC}}^{-1}(k)\mathbf{J}(k)^{\top}(\mathbf{u}(k) - \mathbf{y})\right\rangle ds \\
&= \underbrace{-\int_{s=0}^{1}\left\langle \frac{\partial\mathbf{u}\left(\mathbf{w}(k)\right)}{\partial\mathbf{w}^{\top}}, \eta\mathbf{F}_{\mathrm{K-FAC}}^{-1}(k)\mathbf{J}(k)^{\top}(\mathbf{u}(k) - \mathbf{y})\right\rangle ds}_{\textcircled{1}} \\
&\quad + \underbrace{\int_{s=0}^{1}\left\langle \frac{\partial\mathbf{u}\left(\mathbf{w}(k)\right)}{\partial\mathbf{w}^{\top}} - \frac{\partial\mathbf{u}\left(\mathbf{w}(s)\right)}{\partial\mathbf{w}^{\top}}, \eta\mathbf{F}_{\mathrm{K-FAC}}^{-1}(k)\mathbf{J}(k)^{\top}(\mathbf{u}(k) - \mathbf{y})\right\rangle ds}_{\textcircled{2}}
\end{aligned}
\tag{48}
$$

We first analyze the first term $\textcircled{1}$ by expanding $\mathbf{F}_{\mathrm{K-FAC}}(k)$:

$$
\begin{aligned}
\textcircled{1} &= \eta\left(\mathbf{X} * \mathbf{S}\right)\left((\mathbf{X}^{\top}\mathbf{X})^{-1} \otimes (\mathbf{S}^{\top}\mathbf{S})^{-1}\right)\left(\mathbf{X}^{\top} \star \mathbf{S}^{\top}\right)(\mathbf{y} - \mathbf{u}(k)) \\
&= \eta\left(\mathbf{X}(\mathbf{X}^{\top}\mathbf{X})^{-1}\mathbf{X}^{\top} \odot \mathbf{S}(\mathbf{S}^{\top}\mathbf{S})^{-1}\mathbf{S}^{\top}\right)(\mathbf{y} - \mathbf{u}(k)) \\
&= \eta\left(\mathbf{X}(\mathbf{X}^{\top}\mathbf{X})^{-1}\mathbf{X}^{\top} \odot \mathbf{I}\right)(\mathbf{y} - \mathbf{u}(k))
\end{aligned}
\tag{49}
$$

The first equality we used properties of Khatri-Rao, Hadamard and Kronecker products while the second equality used the generalized inverse.

Next, we need to show $\textcircled{2}$ is small (negligible) compared to $\textcircled{1}$. Similar to exact natural gradient, we first assume two conditions hold. However, the first condition is slightly different in the sense that we assume the positive definiteness of $\mathbf{SS}^{\top}$ (instead of $\mathbf{G}$). We also need a stronger version of Condition 2 in the sense that the Jacobian matrix is stable for two consecutive steps $\left\|\int_{s=0}^{1}\mathbf{J}(\mathbf{w}(s)) - \mathbf{J}(\mathbf{w}(k))ds\right\|_{2} \leq \sqrt{\lambda_{\min}(\mathbf{SS}^{\top})}\frac{\lambda_{\min}(\mathbf{XX}^{\top})}{\lambda_{\max}(\mathbf{XX}^{\top})}$. With these two conditions, we are ready to bound the term $\textcircled{2}$.

$$
\begin{aligned}
\textcircled{2} &\leq \eta\sqrt{\lambda_{\min}(\mathbf{SS}^{\top})}\frac{\lambda_{\min}(\mathbf{XX}^{\top})}{\lambda_{\max}(\mathbf{XX}^{\top})}\left((\mathbf{X}^{\top}\mathbf{X}^{\top})^{-1} \otimes (\mathbf{S}^{\top}\mathbf{S})^{-1}\right)\left(\mathbf{X}^{\top} \star \mathbf{S}^{\top}\right)(\mathbf{y} - \mathbf{u}(k)) \\
&= \eta\sqrt{\lambda_{\min}(\mathbf{SS}^{\top})}\frac{\lambda_{\min}(\mathbf{XX}^{\top})}{\lambda_{\max}(\mathbf{XX}^{\top})}\left((\mathbf{X}^{\top}\mathbf{X}^{\top})^{-1}\mathbf{X}^{\top} \star (\mathbf{S}^{\top}\mathbf{S})^{-1}\mathbf{S}^{\top}\right)(\mathbf{y} - \mathbf{u}(k)) \\
&= \eta\sqrt{\lambda_{\min}(\mathbf{SS}^{\top})}\frac{\lambda_{\min}(\mathbf{XX}^{\top})}{\lambda_{\max}(\mathbf{XX}^{\top})}\left((\mathbf{X}^{\top}\mathbf{X}^{\top})^{-1}\mathbf{X}^{\top} \star \mathbf{S}^{\top}(\mathbf{SS}^{\top})^{-1}\right)(\mathbf{y} - \mathbf{u}(k))
\end{aligned}
\tag{50}
$$

The key of bounding $\textcircled{2}$ is to analyze $(\mathbf{X}^{\top}\mathbf{X}^{\top})^{-1}\mathbf{X}^{\top} \star \mathbf{S}^{\top}(\mathbf{SS}^{\top})^{-1}$. For convenience, we denote this term as $\textcircled{3}$. By the identity of $(\mathbf{A} * \mathbf{B})(\mathbf{A}^{\top} \star \mathbf{B}^{\top}) = \mathbf{AA}^{\top} \odot \mathbf{BB}^{\top}$, we have

$$
\begin{aligned}
\sigma_{\max}(\textcircled{3}) &= \sqrt{\lambda_{\max}(\textcircled{3}^{\top}\textcircled{3})} \\
&= \sqrt{\lambda_{\max}\left(\mathbf{X}(\mathbf{X}^{\top}\mathbf{X})^{-1}(\mathbf{X}^{\top}\mathbf{X})^{-1}\mathbf{X}^{\top} \odot (\mathbf{SS}^{\top})^{-1}\right)}
\end{aligned}
\tag{51}
$$

According to Lemma 1, we have

$$
\sigma_{\max}\left(\textcircled{3}\right) \leq \sqrt{\frac{1}{\lambda_{\min}(\mathbf{X}^{\top}\mathbf{X})^{2}}\frac{1}{\lambda_{\min}(\mathbf{SS}^{\top})}}
\tag{52}
$$

Also, we have

$$
\lambda_{\min}\left(\mathbf{X}(\mathbf{X}^{\top}\mathbf{X})^{-1}\mathbf{X}^{\top} \odot \mathbf{I}\right) \geq \frac{1}{\lambda_{\max}(\mathbf{X}^{\top}\mathbf{X})}
\tag{53}
$$

By choosing a slightly larger $m$, we can show that $\textcircled{1} \gg \textcircled{2}$. Therefore, we can safely ignore $\textcircled{2}$ in eqn. (48) and get

$$
\begin{aligned}
\|\mathbf{y} - \mathbf{u}(k+1)\|_2^2 &= \|\mathbf{y} - \mathbf{u}(k) - (\mathbf{u}(k+1) - \mathbf{u}(k))\|_2^2 \\
&= \|\mathbf{y} - \mathbf{u}(k)\|_2^2 - 2(\mathbf{y} - \mathbf{u}(k))^\top (\mathbf{u}(k+1) - \mathbf{u}(k)) + \|\mathbf{u}(k+1) - \mathbf{u}(k)\|_2^2 \\
&\approx \|\mathbf{y} - \mathbf{u}(k)\|_2^2 - 2\eta \left(\mathbf{y} - \mathbf{u}(k)\right)^\top \left(\mathbf{X}(\mathbf{X}^\top \mathbf{X})^{-1}\mathbf{X}^\top \odot \mathbf{I}\right)(\mathbf{y} - \mathbf{u}(k)) \\
&\quad + \eta^2 \left(\mathbf{y} - \mathbf{u}(k)\right)^\top \left(\mathbf{X}(\mathbf{X}^\top \mathbf{X})^{-1}\mathbf{X}^\top \odot \mathbf{I}\right)^2 (\mathbf{y} - \mathbf{u}(k)) \\
&\leq \|\mathbf{y} - \mathbf{u}(k)\|_2^2 - \eta \left(\mathbf{y} - \mathbf{u}(k)\right)^\top \left(\mathbf{X}(\mathbf{X}^\top \mathbf{X})^{-1}\mathbf{X}^\top \odot \mathbf{I}\right)(\mathbf{y} - \mathbf{u}(k)) \\
&\leq \left(1 - \frac{\eta}{\lambda_{\max}(\mathbf{X}^\top \mathbf{X})}\right)\|\mathbf{y} - \mathbf{u}(k)\|_2^2.
\end{aligned}
$$

(54)

In the last second inequality we used the fact that $\lambda_{\max}\left(\mathbf{X}(\mathbf{X}^\top \mathbf{X})^{-1}\mathbf{X}^\top \odot \mathbf{I}\right) \leq \frac{1}{\lambda_{\min}(\mathbf{X}^\top \mathbf{X})}$ and the step size $\eta = \mathcal{O}\left(\lambda_{\min}\left(\mathbf{X}^\top \mathbf{X}\right)\right)$.

Next, we move on to show the weights of the network remain close to the initialization point.

$$
\begin{aligned}
\|\mathbf{w}(k+1) - \mathbf{w}(0)\|_2 &\leq \eta \sum_{s=0}^{k} \left\| \left((\mathbf{X}^\top \mathbf{X}^\top)^{-1}\mathbf{X}^\top \star \mathbf{S}(s)^\top (\mathbf{S}(s)\mathbf{S}(s)^\top)^{-1}\right)(\mathbf{y} - \mathbf{u}(k))\right\|_2 \\
&\leq \eta \sum_{s=0}^{k} \frac{\|\mathbf{y} - \mathbf{u}(k)\|_2}{\sqrt{\lambda_{\mathbf{S}}/2}} \frac{1}{\lambda_{\min}\left(\mathbf{X}^\top \mathbf{X}\right)} \\
&\leq \eta \sum_{s=0}^{k}(1 - \frac{\eta}{\lambda_{\max}\left(\mathbf{X}^\top \mathbf{X}\right)})^{s/2} \frac{\|\mathbf{y} - \mathbf{u}(0)\|_2}{\sqrt{\lambda_{\mathbf{S}}/2}} \frac{1}{\lambda_{\min}\left(\mathbf{X}^\top \mathbf{X}\right)} \\
&\leq \frac{2\|\mathbf{y} - \mathbf{u}(0)\|_2}{\sqrt{\lambda_{\mathbf{S}}/2}} \frac{\lambda_{\max}\left(\mathbf{X}^\top \mathbf{X}\right)}{\lambda_{\min}\left(\mathbf{X}^\top \mathbf{X}\right)}
\end{aligned}
$$

(55)

Based on matrix pertubation analysis (similar to Lemma 7), it is easy to show that if $m = \mathcal{O}\left(\frac{n^4}{\nu^2 \lambda_{\mathbf{S}}^4 \kappa_{\mathbf{x}^\top \mathbf{x}}^4 \delta^3}\right)$ and step size $\eta = \mathcal{O}\left(\lambda_{\min}\left(\mathbf{X}^\top \mathbf{X}\right)\right)$, we have

$$
\lambda_{\min}\left(\mathbf{S}\mathbf{S}^\top\right) \geq \frac{\lambda_{\mathbf{S}}}{2} \text{ and } \left\|\int_{s=0}^{1} \mathbf{J}(\mathbf{w}(s)) - \mathbf{J}(\mathbf{w}(k))ds\right\|_2 \leq \sqrt{\lambda_{\min}(\mathbf{S}\mathbf{S}^\top)}\frac{\lambda_{\min}(\mathbf{X}\mathbf{X}^\top)}{\lambda_{\max}(\mathbf{X}\mathbf{X}^\top)}
$$

(56)

### D.3 Proof of Positive Definiteness of $\mathbf{S}^\infty \mathbf{S}^{\infty \top}$

Following Du et al. [2018b]'s proof that $\mathbf{G}^\infty$ is strictly positive definite, we apply the same argument to $\mathbf{S}^\infty \mathbf{S}^{\infty \top}$. Recall that $\mathbf{S} = [\phi'(\mathbf{X}\mathbf{w}_1), ..., \phi'(\mathbf{X}\mathbf{w}_m)]$ which is the pre-activation derivative. In our setting, $\phi'(\mathbf{w}^\top \mathbf{x}) = \mathbb{I}\{\mathbf{w}^\top \mathbf{x} \geq 0\}$. For each $\mathbf{x}_i \in \mathbb{R}^d$, it induces an infinite-dimensional feature map $\phi'(\mathbf{x}_i) \in \mathcal{H}$, where $\mathcal{H}$ is the Hilbert space of integrable $d$-dimensional vector fields on $\mathbb{R}^d$.

Now to prove that $\mathbf{S}^\infty \mathbf{S}^{\infty \top}$ is strictly positive definite, it is equivalent to show $\phi'(\mathbf{x}_1), ..., \phi'(\mathbf{x}_n) \in \mathcal{H}$ are linearly independent. Suppose that there are $\alpha_1, ..., \alpha_n \in \mathbb{R}$ such that

$$
\alpha_1 \phi'(\mathbf{x}_1) + ... + \alpha_n \phi'(\mathbf{x}_n) = 0
$$

(57)

We now prove that $\alpha_i = 0$ for all $i$.

We define $D_i = \left\{\mathbf{w} \in \mathbb{R}^d : \mathbf{w}^\top \mathbf{x}_i = 0\right\}$. As shown by Du et al. [2018b], $D_i \not\subset \cup_{j \neq i} D_j$. For a fixed $i \in [n]$, we can choose $\mathbf{z} \in D_i \setminus \cup_{j \neq i} D_j$. We can pick a small enough radius $r_0 > 0$ such that $B(\mathbf{z}, r) \cap D_j = \emptyset, \forall j \neq i, r \leq r_0$. Let $B(\mathbf{z}, r) = B_r^+ \cup B_r^-$, where $B_r^+ = B(\mathbf{z}, r) \cap D_i$.

For $j \neq i$, $\phi'(\mathbf{x}_j)$ is continuous in the neighborhood of $\mathbf{z}$, therefore for any $\epsilon > 0$ there is a small enough $r$ such that

$$
\forall \mathbf{w} \in B(\mathbf{z}, r), \phi'(\mathbf{w}^\top \mathbf{x}_j) = \phi'(\mathbf{z}^\top \mathbf{x}_j)
$$

(58)

Let $\mu$ be Lebesgue measure on $\mathbb{R}^d$, we have

$$
\lim_{r \to 0^+} \frac{1}{\mu(B_r^+)} \int_{B_r^+} \phi'(\mathbf{w}^\top \mathbf{x}_j)d\mathbf{w} = \lim_{r \to 0^+} \frac{1}{\mu(B_r^-)} \int_{B_r^-} \phi'(\mathbf{w}^\top \mathbf{x}_j)d\mathbf{w} = \phi'(\mathbf{z}^\top \mathbf{x}_j)
$$

(59)

Now recall that $\sum_i \alpha_i \phi'(x_i) \equiv 0$, we have

$$
\begin{aligned}
0 &= \lim_{r \to 0^+} \frac{1}{\mu(B_r^+)} \int_{B_r^+} \sum_j \alpha_j \phi'(\mathbf{w}^\top \mathbf{x}_j) d\mathbf{w} - \lim_{r \to 0^+} \frac{1}{\mu(B_r^-)} \int_{B_r^-} \sum_j \alpha_j \phi'(\mathbf{w}^\top \mathbf{x}_j) d\mathbf{w} \\
&= \sum_j \alpha_j \left( \lim_{r \to 0^+} \frac{1}{\mu(B_r^+)} \int_{B_r^+} \phi'(\mathbf{w}^\top \mathbf{x}_j) d\mathbf{w} - \lim_{r \to 0^+} \frac{1}{\mu(B_r^-)} \int_{B_r^-} \phi'(\mathbf{w}^\top \mathbf{x}_j) d\mathbf{w} \right) \qquad (60) \\
&= \sum_j \alpha_j \delta_{ij} = \alpha_i
\end{aligned}
$$

where $\delta_{ij}$ is the Kronecker delta. We complete the proof.

### D.4  Proof of Theorem 5

It has been shown that the Gram matrix for infinite width networks has the following form [Xie et al., 2016, Arora et al., 2019b]:

$$
\mathbf{G}_{ij}^\infty = \left( \frac{\pi - \arccos(\mathbf{x}_i^\top \mathbf{x}_j)}{2\pi} \right) \mathbf{x}_i^\top \mathbf{x}_j \triangleq h(\mathbf{x}_i, \mathbf{x}_j) \qquad (61)
$$

We note that any function defined on the unit sphere has a spherical harmonic decomposition:

$$
h(\mathbf{x}_i, \mathbf{x}_j) = \sum_{u=1}^\infty \gamma_u \phi(\mathbf{x}_i) \phi(\mathbf{x}_j) \qquad (62)
$$

where $\phi_u(\mathbf{x}) : \mathbb{S}^{d-1} \to \mathbb{R}$ is a spherical harmonic basis. In eqn. (62), we sort the spectrum $\gamma_u$ by magnitude. As shown by Xie et al. [2016], $\gamma_n = \Omega\left(n^{\beta-1}\right)$, where $\beta \in (0, 0.5)$. Then we can also define the truncated function:

$$
\left[\mathbf{G}_{ij}^\infty\right]^n = h^n(\mathbf{x}_i, \mathbf{x}_j) = \sum_{u=1}^n \gamma_u \phi(\mathbf{x}_i) \phi(\mathbf{x}_j) \qquad (63)
$$

Due to the fact that $\mathbf{G}^\infty - [\mathbf{G}^\infty]^n$ is PSD, we can then bound the smallest eigenvalue of $[\mathbf{G}^\infty]^n$. Define a matrix $\mathbf{K} \in \mathbb{R}^{n \times n}$ whose rows are

$$
\mathbf{K}^i = [\sqrt{\gamma_1}\phi_1(\mathbf{x}_i), ..., \sqrt{\gamma_n}\phi_n(\mathbf{x}_i)] \qquad (64)
$$

for $1 \le i \le n$. It is easy to show $[\mathbf{G}^\infty]^n = \mathbf{K}\mathbf{K}^\top$ and $\lambda_{\min}(\mathbf{K}\mathbf{K}^\top) = \lambda_{\min}(\mathbf{K}^\top \mathbf{K})$. So we only need to bound $\lambda_{\min}(\mathbf{K}^\top \mathbf{K})$. Observe that $\mathbf{K}^\top \mathbf{K}$ is the sum of $n$ random matrices $\mathbf{K}^{i\top} \mathbf{K}^i$. Based on the properties of spherical harmonic basis $\mathbb{E}[\phi_i(\mathbf{x})\phi_j(\mathbf{x})] = \delta_{ij}$, we have $\lambda_{\min}\left(\mathbb{E}\left[\mathbf{K}^{i\top} \mathbf{K}^i\right]\right) = n\gamma_n$. Observe that all random matrices $\mathbf{K}^\top \mathbf{K}$ are PSD and upper-bounded in the sense

$$
\|\mathbf{K}^{i\top} \mathbf{K}^i\|_2 \le \operatorname{tr}(\mathbf{K}^{i\top} \mathbf{K}^i) = \sum_{u=1}^n \gamma_u \phi_u(\mathbf{x}_i)^2 \le h(\mathbf{x}_i, \mathbf{x}_i) = \frac{1}{2} \qquad (65)
$$

Therefore, the matrix Chernoff bound gives

$$
\mathbb{P}\left[\lambda_{\min}([\mathbf{G}^\infty]^n) \le (1 - \frac{1}{2})\lambda_{\min}\left(\mathbb{E}\left[\mathbf{K}^\top \mathbf{K}\right]\right)\right] \le n \exp\left(-\frac{1}{2^2}\lambda_{\min}\left(\mathbb{E}\left[\mathbf{K}^\top \mathbf{K}\right]\right)\right) \qquad (66)
$$

According to Xie et al. [2016], $\gamma_n$ decays slower than $\mathcal{O}\left(1/n\right)$. This completes the proof.

## E  Proofs for Section 5

### E.1  Proof of Theorem 6

Similar to Neyshabur et al. [2019], Arora et al. [2019b], we analyze the generalization error based on Rademacher Complexity [Bartlett and Mendelson, 2002]. Based on Rademacher complexity, we have the following generalization bound.

**Lemma 9** (Generalization bound with Rademacher Complexity [Mohri et al., 2018]). *Suppose the loss function $\ell(\cdot, \cdot)$ is bounded in $[0, c]$ and is $\rho$-Lipschitz in the first argument. Then with probability at least $1 - \delta$ over sample $\mathcal{S}$ of size $n$:*

$$\sup_{f \in \mathcal{F}} \{\mathcal{L}_{\mathcal{D}}(f) - \mathcal{L}_{\mathcal{S}}(f)\} \leq 2\rho \mathcal{R}_{\mathcal{S}}(\mathcal{F}) + 3c\sqrt{\frac{\log(2/\delta)}{2n}} \tag{67}$$

Therefore, to get the generalization bound, we only need to calculate the Rademacher complexity of a certain function class. Lemma 3 suggests that the learned function $f(\mathbf{w}, \mathbf{a})$ from NGD is in a restricted class of neural networks whose weights are close to the initialization $\mathbf{w}(0)$. The following lemma bounds the Rademacher complexity of this function class.

**Lemma 10** (Rademacher Complexity for a Restricted Class Neural Nets [Arora et al., 2019b]). *For given $A, B > 0$, with probability at least $1 - \delta$ over the random initialization, the following function class*

$$\mathcal{F}_{A,B} = \{f(\mathbf{w}, \mathbf{a}) : \forall r \in [m], \|\mathbf{w}_r - \mathbf{w}_r(0)\|_2 \leq A, \|\mathbf{w} - \mathbf{w}(0)\|_2 \leq B\} \tag{68}$$

*has empirical Rademacher complexity bounded as*

$$\mathcal{R}_{\mathcal{S}}(\mathcal{F}_{A,B}) \leq \frac{B}{\sqrt{2n}}\left(1 + \left(\frac{2\log\frac{2}{\delta}}{m}\right)^{1/4}\right) + 2A^2\sqrt{m} + A\sqrt{2\log\frac{2}{\delta}} \tag{69}$$

With Lemma 9 and 10 at hand, we are only left to bound the distance of the weights to their initialization. Recall the update rule for $\mathbf{w}$:

$$\mathbf{w}(k+1) = \mathbf{w}(k) - \eta \mathbf{J}(k)^\top \mathbf{G}(k)^{-1}(\mathbf{u}(k) - \mathbf{y}) \tag{70}$$

The following lemma upper bounds the distance for each hidden unit.

**Lemma 11.** *If two conditions hold for $s = 0, ..., k$, then we have*

$$\|\mathbf{w}_r(k+1) - \mathbf{w}_r(0)\|_2 \leq \frac{4\sqrt{n}\|\mathbf{y} - \mathbf{u}(0)\|_2}{\sqrt{m}\lambda_0} = \mathcal{O}\left(\frac{n}{\lambda_0 m^{1/2}\delta^{1/2}}\right) \tag{71}$$

To analyze the whole weight vector, we start with ideal case – infinite width network. In that case, both $\mathbf{J}$ and $\mathbf{G}$ are constant matrix throughout the training, and the function error decay exponentially. It is easy to show that the distance is given by

$$\|\mathbf{w}(\infty) - \mathbf{w}(0)\|_2 = \sqrt{(\mathbf{y} - \mathbf{u}(0))^\top (\mathbf{G}^\infty)^{-1}(\mathbf{y} - \mathbf{u}(0))} \tag{72}$$

In the case of finite wide networks, $\mathbf{J}$ and $\mathbf{G}$ would change along the weights, but we can bound the changes and show the norms are small if the network are wide enough. Therefore, the distance is dominated by eqn. (72). From Lemma 3, we know $\|\mathbf{w}(k+1) - \mathbf{w}(0)\|_2 = \mathcal{O}\left(\sqrt{\frac{n}{\lambda_0\delta}}\right)$. According to Lemma 7, it is easy to show that $\|\mathbf{J}(k) - \mathbf{J}(0)\|_2 = \mathcal{O}\left(\frac{n^{2/3}}{\nu^{1/3}\lambda_0^{1/6}m^{1/6}\delta^{1/2}}\right)$ and $\|\mathbf{G}(k) - \mathbf{G}(0)\|_2 = \mathcal{O}\left(\frac{n^{4/3}}{\nu^{2/3}\lambda_0^{1/3}m^{1/3}\delta}\right)$. With these bounds at hand, we are ready to bound $\|\mathbf{w}(\infty) - \mathbf{w}(0)\|_2$ for finite wide networks.

**Lemma 12.** *Under the same setting as Theorem 3, with probability at least $1 - \delta$ over the random initialization, we have*

$$\|\mathbf{w} - \mathbf{w}(0)\|_2 \leq \sqrt{\mathbf{y}^\top(\mathbf{G}^\infty)^{-1}\mathbf{y}} + \mathcal{O}\left(\sqrt{\frac{n}{\lambda_0\delta}}\nu\right) + \frac{\text{poly}(n, \frac{1}{\lambda_0}, \frac{1}{\delta})}{m^{1/4}} \tag{73}$$

Finally, we know that for any sample $\mathcal{S}$ drawn from data distribution $\mathcal{D}$, with probability at least $1 - \delta/3$ over the random initialization, the following hold simultaneously:

1. Optimization succeeds (Theorem 3):

$$\|\mathbf{u}(k) - \mathbf{y}\|_2^2 \leq (1 - \eta)^k \cdot \mathcal{O}\left(\frac{n}{\delta}\right) \leq \frac{n\epsilon^2}{4} \tag{74}$$

This implies an upper bound on the training error:

$$\mathcal{L}_S(f) = \frac{1}{n} \sum_{i=1}^{n} [\ell(u_i(k), y_i)] \leq \frac{1}{n} \sum_{i=1}^{n} |u_i(k) - y_i|$$

$$\leq \frac{1}{\sqrt{n}} \|\mathbf{u}(k) - \mathbf{y}\|_2 = \frac{\epsilon}{2} \tag{75}$$

2. The learned function $f(\mathbf{w}, \mathbf{a})$ belongs to the restricted function class (68).

3. The function class $\mathcal{F}_{A,B}$ has Rademacher complexity bounded as

$$\mathcal{R}_S(\mathcal{F}_{A,B}) \leq \sqrt{\frac{\mathbf{y}^{\top}(\mathbf{G}^{\infty})^{-1}\mathbf{y}}{2n}} + \mathcal{O}\left(\frac{\nu}{\sqrt{\lambda_0 \delta}}\right) + \frac{\text{poly}(n, \frac{1}{\lambda_0}, \frac{1}{\delta})}{\nu^{1/2} m^{1/4}}$$

$$= \sqrt{\frac{\mathbf{y}^{\top}(\mathbf{G}^{\infty})^{-1}\mathbf{y}}{2n}} + \frac{\epsilon}{4} \tag{76}$$

Also, with the probability at least $1 - \delta/3$ over the sample $\mathcal{S}$, we have

$$\sup_{f \in \mathcal{F}} \{\mathcal{L}_{\mathcal{D}}(f) - \mathcal{L}_S(f)\} \leq 2\mathcal{R}_S(\mathcal{F}) + 3\sqrt{\frac{\log(6/\delta)}{2n}} \tag{77}$$

Taking a union bound, we have know that with probability at least $1 - \frac{2}{3}\delta$ over the sample $\mathcal{S}$ and the random initialization, we have

$$\sup_{f \in \mathcal{F}} \{\mathcal{L}_{\mathcal{D}}(f) - \mathcal{L}_S(f)\} \leq \sqrt{\frac{2\mathbf{y}^{\top}(\mathbf{G}^{\infty})^{-1}\mathbf{y}}{n}} + 3\sqrt{\frac{\log(6/\delta)}{2n}} + \frac{\epsilon}{2} \tag{78}$$

which implies

$$\mathcal{L}_{\mathcal{D}}(f) \leq \sqrt{\frac{2\mathbf{y}^{\top}(\mathbf{G}^{\infty})^{-1}\mathbf{y}}{n}} + 3\sqrt{\frac{\log(6/\delta)}{2n}} + \epsilon \tag{79}$$

### E.2 Technical Proofs for Generalization Analysis

*Proof of Lemma 11.* From Theorem 3, we have

$$\|\mathbf{y} - \mathbf{u}(k)\|_2 \leq \sqrt{(1-\eta)^k} \|\mathbf{y} - \mathbf{u}(0)\|_2 \leq (1 - \frac{\eta}{2})^k \|\mathbf{y} - \mathbf{u}(0)\|_2 \tag{80}$$

Recall the natural gradient update rule:

$$\mathbf{w}(k+1) = \mathbf{w}(k) - \eta \mathbf{J}(k)^{\top} \mathbf{G}(k)^{-1}(\mathbf{u}(k) - \mathbf{y}) \tag{81}$$

By $\lambda_{\min}(\mathbf{G}(k)) \geq \frac{\lambda_0}{2}$, we have $\|\mathbf{G}(k)^{-1}(\mathbf{u}(k) - \mathbf{y})\|_2 \leq \frac{2}{\lambda_0} \|\mathbf{u}(k) - \mathbf{y}\|_2$ and then

$$\|\mathbf{w}_r(k+1) - \mathbf{w}_r(k)\|_2 \leq \frac{2\eta\sqrt{n}}{\sqrt{m}\lambda_0} \|\mathbf{u}(k) - \mathbf{y}\|_2 \tag{82}$$

Therefore, we have

$$\|\mathbf{w}_r(k) - \mathbf{w}_r(0)\|_2 \leq \sum_{s=0}^{k-1} \|\mathbf{w}_r(s+1) - \mathbf{w}_r(s)\|_2 \leq \sum_{s=0}^{k-1} \frac{2\eta\sqrt{n}}{\sqrt{m}\lambda_0} \|\mathbf{u}(s) - \mathbf{y}\|_2$$

$$\leq \frac{2\eta\sqrt{n}}{\sqrt{m}\lambda_0} \sum_{s=0}^{k-1} (1 - \frac{\eta}{2})^s \|\mathbf{u}(0) - \mathbf{y}\|_2 \tag{83}$$

$$\leq \frac{2\eta\sqrt{n}}{\sqrt{m}\lambda_0} \frac{2}{\eta} \|\mathbf{u}(0) - \mathbf{y}\|_2 = \mathcal{O}\left(\frac{n}{\lambda_0 m^{1/2} \delta^{1/2}}\right)$$

$\square$

*Proof of Lemma 12.* To bound the norm of distance, we first decompose the total distance into the sum of each weight update.

$$
\begin{aligned}
\mathbf{w}(k) - \mathbf{w}(0) &= \sum_{s=0}^{k-1} \left( \mathbf{w}(s+1) - \mathbf{w}(s) \right) \\
&= -\sum_{s=0}^{k-1} \eta \mathbf{J}(s)^\top \mathbf{G}(s)^{-1} (\mathbf{u}(s) - \mathbf{y})
\end{aligned}
\tag{84}
$$

We then analyze the term $\mathbf{u}(k) - \mathbf{y}$, which evolves as follows.

**Lemma 13.** *Under the same setting as Theorem 3, with probability at least $1 - \delta$ over the random initialization, we have*

$$
\mathbf{u}(k) - \mathbf{y} = -(1-\eta)^k \mathbf{y} + \zeta(k)
\tag{85}
$$

*where* $\|\zeta(k)\|_2 = \mathcal{O}\left( (1-\eta)^k \sqrt{\frac{n}{\delta}} \nu + k \left(1 - \frac{\eta}{2}\right)^{k-1} \eta \frac{n^{7/6}}{\nu^{1/3}\lambda_0^{2/3}m^{1/6}\delta} \right)$

Plugging eqn. (85) into eqn. (84), we have

$$
\mathbf{w}(k) - \mathbf{w}(0) = \sum_{s=0}^{k-1} \eta \mathbf{J}(s)^\top \mathbf{G}(s)^{-1}(1-\eta)^s \mathbf{y} - \eta \mathbf{J}(s)^\top \mathbf{G}(s)^{-1}\zeta(s)
\tag{86}
$$

The RHS term in above equation is considered perturbation and we can upper bound their norm easily. By Lemma 13, we have

$$
\|\mathbf{J}(s)^\top \mathbf{G}(s)^{-1}\zeta(s)\|_2 = \mathcal{O}\left( (1-\eta)^s \sqrt{\frac{n}{\lambda_0 \delta}} \nu + s \left(1 - \frac{\eta}{2}\right)^{s-1} \eta \frac{n^{7/6}}{\nu^{1/3}\lambda_0^{7/6}m^{1/6}\delta} \right)
\tag{87}
$$

Plugging eqn. (87) into eqn. (84), we have

$$
\begin{aligned}
\mathbf{w}(k) - \mathbf{w}(0) &= \sum_{s=0}^{k-1} \eta \mathbf{J}(s)^\top \mathbf{G}(s)^{-1}(1-\eta)^s \mathbf{y} + \mathbf{e}_1 \\
&= \sum_{s=0}^{k-1} \eta \mathbf{J}(s)^\top (\mathbf{G}^\infty)^{-1}(1-\eta)^s \mathbf{y} + \mathbf{e}_1 + \mathbf{e}_2 \\
&= \sum_{s=0}^{k-1} \eta \mathbf{J}(0)^\top (\mathbf{G}^\infty)^{-1}(1-\eta)^s \mathbf{y} + \mathbf{e}_1 + \mathbf{e}_2 + \mathbf{e}_3
\end{aligned}
\tag{88}
$$

For $\mathbf{e}_1$, we have $\|\mathbf{e}_1\|_2 = \mathcal{O}\left( \sqrt{\frac{n}{\lambda_0 \delta}} \nu + \frac{n^{7/6}}{\nu^{1/3}\lambda_0^{7/6}m^{1/6}\delta} \right)$ by using the following inequality:

$$
\sum_{s=0}^{k-1} s \left(1 - \frac{\eta}{2}\right)^{s-1} \leq \sum_{s=0}^{\infty} s \left(1 - \frac{\eta}{2}\right)^{s-1} = \frac{4}{\eta^2}
\tag{89}
$$

For $\mathbf{e}_2$, we have

$$
\|\mathbf{e}_2\|_2 \leq \sum_{s=0}^{k-1} \eta(1-\eta)^s \|\mathbf{J}(s)\|_2 \|\mathbf{G}(s)^{-1} - (\mathbf{G}^\infty)^{-1}\|_2 \|\mathbf{y}\|_2
\tag{90}
$$

In eqn. (90), we need first bound $\|\mathbf{G}(s)^{-1} - (\mathbf{G}^\infty)^{-1}\|_2 \|\mathbf{y}\|_2$. The following lemma bounds this norm by expanding the inverse with an infinite series.

**Lemma 14.** *Under the same setting as Theorem 3, with probability at least $1 - \delta$ over the random initialization, we have*

$$
\|\mathbf{G}(k)^{-1} - \mathbf{G}(0)^{-1}\|_2 \approx \|\mathbf{G}(k)^{-1} - (\mathbf{G}^\infty)^{-1}\|_2 = \mathcal{O}\left( \frac{n^{4/3}}{\nu^{2/3}\lambda_0^{7/3}m^{1/3}\delta} \right)
\tag{91}
$$

It is easy to show that both $\|\mathbf{J}\|_2$ and $\|\mathbf{y}\|_2$ are $\mathcal{O}\left(\sqrt{n}\right)$. Plugging them back into eqn. (90), we have

$$\|\mathbf{e}_2\|_2 = \mathcal{O}\left(\frac{n^{7/3}}{\nu^{2/3}\lambda_0^{7/3}m^{1/3}\delta}\right) \tag{92}$$

Similarly, we can bound $\mathbf{e}_3$ as follows,

$$\|\mathbf{e}_3\|_2 \le \sum_{s=0}^{k-1}\eta(1-\eta)^s\|\mathbf{J}(s)-\mathbf{J}(0)\|_2\|(\mathbf{G}^\infty)^{-1}\|_2\|\mathbf{y}\|_2 = \mathcal{O}\left(\frac{n^{7/6}}{\nu^{1/3}\lambda_0^{7/6}m^{1/6}\delta^{1/2}}\right) \tag{93}$$

We also bound the first term in eqn. (88):

$$
\begin{aligned}
&\left\|\sum_{s=0}^{k-1}\eta\mathbf{J}(0)^\top(\mathbf{G}^\infty)^{-1}(1-\eta)^s\mathbf{y}\right\|_2^2 \\
&\le \left\|\mathbf{J}(0)^\top(\mathbf{G}^\infty)^{-1}\mathbf{y}\right\|_2^2 \\
&= \mathbf{y}^\top(\mathbf{G}^\infty)^{-1}\mathbf{J}(0)\mathbf{J}(0)^\top(\mathbf{G}^\infty)^{-1}\mathbf{y} \\
&\le \mathbf{y}^\top(\mathbf{G}^\infty)^{-1}\mathbf{y} + \|\mathbf{G}(0)-\mathbf{G}^\infty\|_2\|(\mathbf{G}^\infty)^{-1}\|_2^2\|\mathbf{y}\|_2^2 \\
&= \mathbf{y}^\top(\mathbf{G}^\infty)^{-1}\mathbf{y} + \mathcal{O}\left(\frac{n^2\sqrt{\log\frac{n}{\delta}}}{\lambda_0^2 m^{1/2}}\right)
\end{aligned}
\tag{94}
$$

Combining bounds (87), (90), (93) and (94), we have

$$
\begin{aligned}
\|\mathbf{w}-\mathbf{w}(0)\|_2 &\le \sqrt{y^\top(\mathbf{G}^\infty)^{-1}\mathbf{y}} + \mathcal{O}\left(\sqrt{\frac{n^2\sqrt{\log\frac{n}{\delta}}}{\lambda_0^2 m^{1/2}}}\right) \\
&\quad + \mathcal{O}\left(\sqrt{\frac{n}{\lambda_0\delta}}\nu + \frac{n^{7/6}}{\nu^{1/3}\lambda_0^{7/6}m^{1/6}\delta}\right) + \mathcal{O}\left(\frac{n^{7/3}}{\nu^{2/3}\lambda_0^{7/3}m^{1/3}\delta}\right) \\
&= \sqrt{\mathbf{y}^\top(\mathbf{G}^\infty)^{-1}\mathbf{y}} + \mathcal{O}\left(\sqrt{\frac{n}{\lambda_0\delta}}\nu\right) + \frac{\operatorname{poly}(n,\frac{1}{\lambda_0},\frac{1}{\delta})}{\nu^{1/3}m^{1/6}}
\end{aligned}
\tag{95}
$$

This finishs the proof. $\qquad\square$

*Proof of Lemma 13.* Recall that in eqn. (24), we have

$$
\begin{aligned}
\mathbf{u}(k+1)-\mathbf{u}(k) &\le \eta\left(\mathbf{y}-\mathbf{u}(k)\right) + \eta\left[\mathbf{J}(k+1)-\mathbf{J}(k)\right]\mathbf{J}(k)^\top\mathbf{G}(k)^{-1}(\mathbf{y}-\mathbf{u}(k)) \\
&= \eta\left(\mathbf{y}-\mathbf{u}(k)\right) + \xi(k)
\end{aligned}
\tag{96}
$$

where $\|\xi(k)\|_2 = \mathcal{O}\left(\eta\left(1-\frac{\eta}{2}\right)^k\frac{n^{7/6}}{\nu^{1/3}\lambda_0^{2/3}m^{1/6}\delta}\right)$. Applying eqn. (96) recursively, we get

$$
\begin{aligned}
\mathbf{u}(k)-\mathbf{y} &= (1-\eta)^k(\mathbf{u}(0)-\mathbf{y}) + \sum_{s=0}^{k-1}(1-\eta)^s\xi(k-1-s) \\
&= -(1-\eta)^k\mathbf{y} + (1-\eta)^k\mathbf{u}(0) + \sum_{s=0}^{k-1}(1-\eta)^s\xi(k-1-s)
\end{aligned}
\tag{97}
$$

For the second term, we have $\|(1-\eta)^k\mathbf{u}(0)\|_2 = \mathcal{O}\left((1-\eta)^k\sqrt{\frac{n}{\delta}}\nu\right)$. For the last term, we have

$$
\begin{aligned}
\left\|\sum_{s=0}^{k-1}(1-\eta)^s\xi(k-1-s)\right\|_2 &\le \sum_{s=0}^{k-1}\eta\left(1-\frac{\eta}{2}\right)^{k-1}\mathcal{O}\left(\frac{n^{7/6}}{\nu^{1/3}\lambda_0^{2/3}m^{1/6}\delta}\right) \\
&= \mathcal{O}\left(k\eta\left(1-\frac{\eta}{2}\right)^{k-1}\frac{n^{7/6}}{\nu^{1/3}\lambda_0^{2/3}m^{1/6}\delta}\right)
\end{aligned}
\tag{98}
$$

$\qquad\square$

*Proof of Lemma 14.* Notice that $\mathbf{G}^{-1} = \alpha \sum_{s=0}^{\infty} (\mathbf{I} - \alpha\mathbf{G})^s$, as long as $\alpha$ is small enough so that $\mathbf{I} - \alpha\mathbf{G}$ is positive definite. Therefore, instead of bounding $\|\mathbf{G}(s)^{-1} - (\mathbf{G}^{\infty})^{-1}\|_2$ directly, we can upper bound the following quantity:

$$\left\|\sum_{s=0}^{\infty} (\mathbf{I} - \alpha\mathbf{G}(k))^s - (\mathbf{I} - \alpha\mathbf{G}^{\infty})^s \right\|_2 \leq \sum_{s=0}^{\infty} \|(\mathbf{I} - \alpha\mathbf{G}(k))^s - (\mathbf{I} - \alpha\mathbf{G}^{\infty})^s\|_2 \qquad (99)$$

Let $e(s)$ denote $\|(\mathbf{I} - \alpha\mathbf{G}(k))^s - (\mathbf{I} - \alpha\mathbf{G}^{\infty})^s\|_2$, we then have the following recursion:

$$e(s+1) \leq \|\mathbf{I} - \alpha\mathbf{G}^{\infty}\|_2\, e(s) + \|\alpha\mathbf{G}(k) - \alpha\mathbf{G}^{\infty}\|_2 \|(\mathbf{I} - \alpha\mathbf{G}(k))^s\|_2 \qquad (100)$$

Recall that $\lambda_{\min}(\mathbf{G}^{\infty}) = \lambda_0 > 0$ and $\lambda_{\min}(\mathbf{G}(k)) \geq \frac{1}{2}\lambda_0$, we have

$$e(s+1) \leq (1 - \alpha\lambda_0)e(s) + \|\alpha\mathbf{G}(k) - \alpha\mathbf{G}^{\infty}\|_2 \left(1 - \frac{1}{2}\alpha\lambda_0\right)^s \qquad (101)$$

Also we can easily bound the deviation of $\mathbf{G}(k)$ from $\mathbf{G}^{\infty}$ as follows,

$$\begin{aligned}
\|\mathbf{G}(k) - \mathbf{G}^{\infty}\|_2 &\leq \|\mathbf{G}(k) - \mathbf{G}(0)\|_2 + \|\mathbf{G}(0) - \mathbf{G}^{\infty}\|_2 \\
&= \mathcal{O}\left(\frac{n^{4/3}}{\nu^{2/3}\lambda_0^{1/3}m^{1/3}\delta}\right) + \mathcal{O}\left(\frac{n\sqrt{\log\frac{n}{\delta}}}{m^{1/2}}\right) \\
&= \mathcal{O}\left(\frac{n^{4/3}}{\nu^{2/3}\lambda_0^{1/3}m^{1/3}\delta}\right) \triangleq E
\end{aligned} \qquad (102)$$

Plugging eqn. (102) into eqn. (101), we have

$$e(s+1) \leq (1 - \alpha\lambda_0)e(s) + \alpha E \left(1 - \frac{1}{2}\alpha\lambda_0\right)^s \qquad (103)$$

With basic techniques of series theory and the fact $e(1) = \alpha E$, we have the following result:

$$e(s) \leq \alpha E \left[\frac{\left(1 - \frac{1}{2}\alpha\lambda_0\right)^s}{\frac{1}{2}\alpha\lambda_0} + \left(2 - \frac{1}{\frac{1}{2}\alpha\lambda_0}\right)(1 - \alpha\lambda_0)^{s-1}\right], \; \forall s \geq 1 \qquad (104)$$

Plugging eqn. (104) back into eqn. (99), we get

$$\begin{aligned}
&\left\|\sum_{s=0}^{\infty}(\mathbf{I} - \alpha\mathbf{G}(k))^s - (\mathbf{I} - \alpha\mathbf{G}^{\infty})^s\right\|_2 \\
&\leq \sum_{s=1}^{\infty} \alpha E \left[\frac{\left(1 - \frac{1}{2}\alpha\lambda_0\right)^s}{\frac{1}{2}\alpha\lambda_0} + \left(2 - \frac{1}{\frac{1}{2}\alpha\lambda_0}\right)(1 - \alpha\lambda_0)^{s-1}\right] \\
&= \frac{2E}{\alpha\lambda_0^2} = \mathcal{O}\left(\frac{n^{4/3}}{\nu^{2/3}\lambda_0^{7/3}m^{1/3}\delta\alpha}\right)
\end{aligned} \qquad (105)$$

and

$$\left\|\mathbf{G}(k)^{-1} - (\mathbf{G}^{\infty})^{-1}\right\|_2 = \mathcal{O}\left(\frac{n^{4/3}}{\nu^{2/3}\lambda_0^{7/3}m^{1/3}\delta}\right) \qquad (106)$$

This completes the proof. □

# F   Asymptotic analysis

As shown by Lee et al. [2019], infinitely wide neural networks are linearized networks in the sense that the first-order Taylor expansion is accurate and the training dynamics of wide neural networks

are well captured by linearized models in practice. Assume linearity (i.e., the Jacobian matrix $\mathbf{J}$ is constant over $\mathbf{w}$), we have the following result:

$$
\begin{aligned}
\mathbf{u}(k+1) - \mathbf{u}(k) &= \mathbf{u}\left(\mathbf{w}(k) - \eta \mathbf{J}(k)^\top \mathbf{G}(k)^{-1}(\mathbf{u}(k) - \mathbf{y})\right) - \mathbf{u}(\mathbf{w}(k)) \\
&= -\int_{s=0}^{\eta} \left\langle \frac{\partial \mathbf{u}\left(\mathbf{w}(k)\right)}{\partial \mathbf{w}^\top}, \mathbf{J}(k)^\top \mathbf{G}(k)^{-1}(\mathbf{u}(k) - \mathbf{y}) \right\rangle ds \\
&\quad + \int_{s=0}^{\eta} \left\langle \underbrace{\frac{\partial \mathbf{u}\left(\mathbf{w}(k)\right)}{\partial \mathbf{w}^\top} - \frac{\partial \mathbf{u}\left(\mathbf{w}(s)\right)}{\partial \mathbf{w}^\top}}_{=\mathbf{0}}, \mathbf{J}(k)^\top \mathbf{G}(k)^{-1}(\mathbf{u}(k) - \mathbf{y}) \right\rangle ds \quad (107) \\
&= -\int_{s=0}^{\eta} \left\langle \mathbf{J}(k), \mathbf{J}(k)^\top \mathbf{G}(k)^{-1}(\mathbf{u}(k) - \mathbf{y}) \right\rangle ds = \eta\left(\mathbf{y} - \mathbf{u}(k)\right)
\end{aligned}
$$

So it is easy to show that

$$
\mathbf{y} - \mathbf{u}(k+1) = (1 - \eta)\left(\mathbf{y} - \mathbf{u}(k)\right) \tag{108}
$$

which means exact natural gradient descent can converge with one iteration if we take $\eta = 1$, demonstrating the effectiveness of natural gradient descent.

Moreover, under the linearized network, we can conveniently analyze the trajectories of GD and NGD. Notably, we analyze the paths taken by GD and NGD in both output space and weight space. The dynamics with infinitesimal step size are summarized as follow.

$$
\begin{aligned}
\frac{d}{dt}\left(\mathbf{y} - \mathbf{u}(t)\right) &= -\mathbf{G}\left(\mathbf{y} - \mathbf{u}(t)\right) & \frac{d}{dt}\left(\mathbf{y} - \mathbf{u}(t)\right) &= -\left(\mathbf{y} - \mathbf{u}(t)\right) \\
\frac{d}{dt}\mathbf{w}_{\mathrm{GD}}(t) &= \mathbf{J}^\top\left(\mathbf{y} - \mathbf{u}(t)\right) & \frac{d}{dt}\mathbf{w}_{\mathrm{NGD}}(t) &= \mathbf{J}^\top \mathbf{G}^{-1}\left(\mathbf{y} - \mathbf{u}(t)\right)
\end{aligned} \tag{109}
$$

By standard matrix differential equation theory, we have

$$
\begin{aligned}
\mathbf{w}_{\mathrm{GD}}(t) &= \mathbf{J}^\top \mathbf{G}^{-1}\left(\mathbf{I} - \exp(-\mathbf{G}t)\right)\left(\mathbf{y} - \mathbf{u}(0)\right) + \mathbf{w}(0) \\
\mathbf{w}_{\mathrm{NGD}}(t) &= (1 - \exp(-t))\,\mathbf{J}^\top \mathbf{G}^{-1}\left(\mathbf{y} - \mathbf{u}(0)\right) + \mathbf{w}(0)
\end{aligned} \tag{110}
$$

By some manipulations, we can show $\mathbf{w}_{\mathrm{GD}}(\infty) = \mathbf{w}_{\mathrm{NGD}}(\infty)$, which means that gradient descent and natural gradient descent converge to the same point, though these two paths are typically different. Notably, the limiting distance $\mathbf{w}(\infty) - \mathbf{w}(0)$ (for both GD and exact NGD) converges to min-norm least squares solution.

## Footnotes

[4]That the gradient of $\ell$ is $L$-Lipschitz implies the gradient of $\mathcal{L}$ is also $L$-Lipschitz.