[Reviews · NeurIPS 2019]

Reviewer 1



After rebuttal: I have carefully read the comments from other reviewers and the feedback from the authors. My main concern was the generalization ability of NGD, but the experiments in the feedback are a bit confused to me because GD doesn't seem to achieve zero training loss but NGD converges to 0 very quickly in MNIST regression. I would suggest the authors provide more details about that experiment setting, e.g., how do you select the hyperparameter. Thus, I would like to keep my score unchanged. ---------------------------------------------------- The authors analyzed the dynamics of Natural Gradient Descent(NGD) with a two-layer over-parametrized ReLU neural network and quadratic loss. The framework for the proof follows the recent line of work about over-parametrization, e.g., the papers written by Du et al, Li and Liang, and Allen-Zhu et al., the core of which is the Gram matrix. The main intuition of this framework is that if the Gram matrix is positive definite, then the gradient will be lower bounded by the minimum eigenvalue of the Gram matrix and the value of objective function (loss function), i.e., the gradient cannot be small unless the objective is small, so we will get a linear convergence rate. Moreover, due to this linear convergence rate, the movement of the weights will be small, making the Gram matrix always positive definite. These two things are in some sense equivalent to the two conditions mentioned in this paper, and the authors proved that with NGD and a two-layer over-parametrized ReLU neural network, one will satisfy these conditions with high probability. The interesting thing about this result is that it achieves a better convergence rate using the second-order optimization algorithm and better tolerance for the step size. The authors also provided analysis for K-FAC in a very similar setting, and still uses a similar framework for the proof. The converge rate and requirement for the width of the network is a bit worse than NGD, but acceptable. The authors analyzed the generalization performance for NGD as well. At several places, the authors made some simplifications for the setting, e.g., fix all the weights on the second layer. This is acceptable because otherwise the movement of the second layer will still be small and doesn't significantly influence the result. This paper is generally well-written and very easy to read. The authors also bounded \lambda_0 in a very simple setting, which is not mentioned in the paper of Du et al., but it would be better to analyze it more carefully. Despite the advantages mentioned above, I have some concerns about this paper, which are listed below: 1. For the generalization result, getting similar generalization bound doesn't necessarily mean that the two algorithms will have a similar ability to generalize. Thus, to make the authors' claim about generalization more convincing, I would suggest the authors do some experiments on some real-world datasets and compare the generalization performance. 2. The authors gave a lower bound for \lambda_0 but doesn't lower bound \lambda_S, so we don't know the difference in width requirements of NGD and K-FAC. The authors can say something about the lower bound of \lambda_S or do some experiments to convince the readers that the requirement of K-FAC is not much worse. 3. The authors are simplifying some settings like fixing the second layer weights, but I think the authors should provide some intuition about why it works for the general setting instead of directly saying "it's easy to ...".

Reviewer 2



This paper studies the convergence performance of natural gradient descent for training over-parameterized neural networks. Specifically, the authors prove that NGD can achieve faster convergence rate and similar generalization performance compared with ordinary gradient descent. Moreover, this paper also proves that K-FAC can also converge to global minima with a linear rate under certain assumptions. This paper is well written and theoretically sound. The limitation of this paper mainly lies in the significance and proof novelty given the existing literature. More specifically, I have the following comments: 1. The studied optimization algorithm is of limited interesting, especially in training deep neural networks. Indeed, the computation of natural gradient is extremely expensive for over-parameterized neural network, since it involves the operations for high-dimensional matrices (matrix inverse and matrix production) in each iteration. Besides, the authors should also clarify why K-FAC can reduce the computation and memory costs compared with the exact calculation of natural gradient, it also involves the inverse calculation for high-dimensional matrices. 2. The proof technique is quite similar to Du et al., 2018b regarding the convergence analysis and Arora et al. 2019b regarding the generalization analysis. Therefore the proof novelty is not significant, and the theoretical results are somehow incremental. 3. In Remark 3, the authors should clearly state the required iteration number to achieve epsilon accuracy, and discuss the comparison with those in Du et al., 2018b, Oymak and Soltanolkotabi, 2019. In addition, the authors should also rigorously illustrate why X^\topX typically has a smaller condition number than the Gram matrix G. 4. In Theorem 6, the initialization of model weights depends on the target accuracy \epsilon, however, in practice the initialization scheme is fixed regardless of the choice of target accuracy. 5. The statement of the 3rd contribution summarized in the introduction is not accurate. Using a larger step size may not be directly related to a faster convergence rate. In Wu et al., 2019, the authors showed that a variant of adaptive gradient descent can also leverage O(1) step size but attain the same convergence rate as gradient descent. 6. The authors should run some simple experiments to verify the theoretical results, including comparison with GD in terms of both iteration number and running time, as did in Bernacchia et al., 2018. 7. In the proof of Theorem 4, condition 2 is verified in (55), which requires that (54) holds. However, to prove (54), condition 2 is necessary. This implies that when verifying condition the authors implicitly assume that such condition holds. The authors should clarify this.

Reviewer 3



Originality: The contribution is original with related work clearly referenced. Quality: The techniques used in the proofs seem sound. It would be interesting to have values for actual networks, for several quantities (see "improvement" part) Clarity: The setup and theorems are clearly presented. Everything is well explained and the proofs are in the appendix. Significance: The natural gradient is a promising way for optimizing and analyzing neural networks. This theoretical contribution for 2 layers ReLU networks is an interesting step toward a more general theory.

[Author Response · NeurIPS 2019]

We thank the reviewers for their detailed reviews and constructive feedback. We will address your questions and
concerns below (due to space constraints, we focus on the main concerns):

**Generalization (R1, R3).** The key insight from the generalization bound is that the generalization gap depends on the
distance of weights to the initialization. In the paper, we have shown that for GD and NGD, the distance they move are
of the same order $\sqrt{n}$ though they take very different paths. Moreover, we showed in Appendix F that GD and NGD
converge to the same min-norm least-square solution in the infinite width limit. The point we intended to make is not
that NGD always generalizes as well as GD, but that our provable generalization bound for NGD is as good as the
known bounds for GD. It is not known how tight any of these bounds are. We will clarify this point in the final version.

**Real data experiments (R1, R2, R3).** Numerous
other papers have compared NGD and SGD on
modern neural net benchmarks in terms of both
convergence and generalization (e.g. see the series
of papers on K-FAC). Here are some additional ex-
periments. Specifically, for regression on MNIST,
we generated the data in the same way as in Figure
1 of the paper, but using 5000 training examples.
For classification, we used the standard training-test
split. For fair comparison, we removed all bells and
whistles (including batch norm, data augmentation,
weight decay). For GD, we don't include the mo-

**Figure 1:** Red lines are GD while blue lines are NGD (Hessian-free). Solid lines are training curves while dashed lines are testing curves. NGD **converges faster** than GD and also **generalizes well**.

mentum since previous theory papers only discussed plain gradient descent. We tuned the learning rate for GD using
standard grid search. For MNIST, we used a two-layer MLP (one hidden layer) with 6000 hidden units. For CIFAR-10,
we used a VGG-style network with 5 conv layers, and the filter count for each layer is $[32, 64, 128, 256, 256]$.

**Stable Jacobian condition (R3).** We have numerical results in our submission. In Figure 1 (page 6), we verified the
stable Jacobian condition on MNIST with 100 training samples. In particular, we showed that for the over-parameterized
network (in the second row), natural gradient descent matches output space gradient descent well (even with a large
learning rate), indicating that the Jacobian is stable enough for the output space path to be nearly linear.

**Removing simplifying assumptions (R1).** The assumption of two-layer networks with a fixed second layer simplifies
the proofs. We believe we can remove these assumptions using the techniques of Du et al., [2018b]. Since almost all
of our analysis is architecture-agnostic, one only needs to check the conditions. Condition 2 was essentially verified
for multi-layer and non-fixed-second-layer networks by Du et al., [2018a]. Intuitively, the conditions just require the
network to behave like a linearized one, and we'd expect this to hold for wide networks of any depth.

**Memory and computation costs of NGD and K-FAC (R2).** Most of our paper analyzes an idealized version of NGD
which practical algorithms like Hessian-free optimization and K-FAC are trying to approximate; hence, it's not intended
to be a practical training procedure. A naïve implementation of exact NGD requires $\mathcal{O}(m^2)$ space to store the Fisher
matrix and $\mathcal{O}(m^3)$ time to invert it (where $m$ is the number of parameters). Expressing the pseudoinverse in terms
of the Gram matrix as we do (see equation (3) in the paper) makes the costs $\mathcal{O}(n^2)$ and $\mathcal{O}(n^3)$, respectively, which
is much smaller for overparameterized networks. We note that this is equivalent to preconditioning the output-space
gradient $\mathbf{u} - \mathbf{y}$ with the Gram matrix, which suggests a *new* way for running natural gradient descent.

K-FAC requires much less memory and computation than exact NGD — in practice, a small constant factor overhead
compared with GD — and has been applied to large modern networks such as ImageNet classifiers [Ba et al., 2018,
Osawa et al., 2018] and large transformers. Specifically, we only need to store and invert small matrices which has
roughly the same shape as weight matrices in each layer. See Martens and Grosse [2015] for detailed discussion.

**Novelty of the proof techniques (R2).** While we borrowed much high-level structure from Du et al.'s analysis, several
aspects of our analysis are novel. First, we significantly improve the bound by *bounding the distance of the whole weight*
*vector*, giving the bound $\Omega(n^4)$. By contrast, the bound in Du et.al., [2018b] and Wu et al., [2019] is $\Omega(n^6)$. Second,
we introduced two modular conditions, making our proofs much clearer and more general than Du et al., [2018b]. Third,
we extend the results to general loss functions in Theorem 2. Lastly, we give an explicit bound for $\lambda_0$.

**Why does a larger step size imply faster convergence? (R2)** A larger step size doesn't imply faster convergence in
general, but it does in the context of Theorem 3 and the analogous result for GD, since the convergence rate is given in
terms of the step size (see lines 244-247 for short discussion). Hence, a larger bound on the step size (in the condition
of the Theorem) implies faster convergence. We'll clarify this in the revised version.

**Proofs for Thm 4 (R2).** Because the proof for K-FAC is simliar to that of NGD, we skipped a step in the proof of Thm
4. We should have included a version of Lemma 4 (see lines 513-521), and will do so in the revision.

[Meta-Review · NeurIPS 2019]

This paper proves fast convergence of natural gradient descent for over-parameterized neural networks, and its generalization error bound. This paper is on the borderline and was carefully discussed. The main concern is about the novelty of this paper, as well as lack of details in the experiments. The paper gathered some support from the reviewers to merit acceptance, after author response and reviewer discussion. Thus I recommend accept.